# Metal-support interaction boosts the stability of Ni-based electrocatalysts for alkaline hydrogen oxidation

Xiaoyu Tian[1,5], Renjie Ren[2,5], Fengyuan Wei[2], Jiajing Pei[3], Zhongbin Zhuang [4], Lin Zhuang [2] ✉ & Wenchao Sheng [1] ✉

Ni-based hydrogen oxidation reaction (HOR) electrocatalysts are promising anode materials for the anion exchange membrane fuel cells (AEMFCs), but their application is hindered by their inherent instability for practical operations. Here, we report a $TiO_2$ supported $Ni_4Mo$ ($Ni_4Mo/TiO_2$) catalyst that can effectively catalyze HOR in alkaline electrolyte with a mass activity of $10.1 \pm 0.9\,A\,g^{-1}_{Ni}$ and remain active even up to 1.2 V. The $Ni_4Mo/TiO_2$ anode AEMFC delivers a peak power density of $520\,mW\,cm^{-2}$ and durability at $400\,mA\,cm^{-2}$ for nearly 100 h. The origin for the enhanced activity and stability is attributed to the down-shifted $d$ band center, caused by the efficient charge transfer from $TiO_2$ to Ni. The modulated electronic structure weakens the binding strength of oxygen species, rendering a high stability. The $Ni_4Mo/TiO_2$ has achieved greatly improved stability both in half cell and single AEMFC tests, and made a step forward for feasibility of efficient and durable AEMFCs.

Hydrogen-oxygen fuel cells with high energy efficiency and durability permit a sustainable energy system based on solar or electrical hydrogen converted from renewable energy resources[1]. With the fast development of low-cost oxygen reduction electrocatalysts at the cathode and alkane chain based anion exchange membranes[2,3], enhancing the efficiency and durability of the anodic hydrogen oxidation reaction (HOR) electrocatalysts becomes particularly vital to the anion exchange membrane fuel cells (AEMFCs), as the HOR activity on the best Pt catalyst is two orders of magnitude slower in alkaline than in acidic medium[4,5]. Higher Pt catalyst loading would thus offset the cost merit brought by the cathode electrocatalysts and the membranes.

Enhancing the HOR activity of electrocatalysts is typically achieved by tuning the electronic structure of materials by alloying Pt with foreign metals. Pt alloys (PtRu, PtFe, PtCo, PtNi and PtCu) show greatly enhanced HOR activity compared to Pt, which is attributed to the optimized hydrogen binding energy (HBE)[6–8] or potentially

together with the optimized surface hydroxyl adsorption energy[9–11]. Similar strategy is also applied to developing non-precious HOR catalysts such as Ni to completely replace Pt group metals. In the few early attempts to improve the HOR activity of Ni, electrodeposited NiMo and CoNiMo thin films show superior activities to Ni, which is attributed to the optimized HBE introduced by the electronic effects between Ni and Mo/CoMo[12]. Recently, a group of Ni-based bimetallic nanoparticles, which are industry-relevant, have been developed for the HOR in base, showing much improved HOR activity with respect to Ni[13,14]. Except for tuning HBE on Ni, researchers also ascribe the improved HOR activity partially to the optimized surface hydroxyl group binding strength on foreign metals, which may facilitate the combination of -H and surface -OH group through the bi-functional mechanism[13]. Despite the debates in understanding the HOR mechanism in base, alloying achieves success in enhancing the overall HOR activities. However, non-precious metal catalysts often fail to meet the stability requirement. For example, the HOR performance on

[1]State Key Laboratory of Pollution Control and Resource Reuse, College of Environmental Science and Engineering, Tongji University, Shanghai Institute of Pollution Control and Ecological Security, Shanghai 200092, PR China. [2]College of Chemistry and Molecular Sciences, Hubei Key Laboratory of Electrochemical Power Sources, Wuhan University, Wuhan 430072, PR China. [3]Beijing Synchrotron Radiation Facility, Institute of High Energy Physics, Chinese Academy of Sciences, Beijing 100049, PR China. [4]State Key Laboratory of Organic-Inorganic Composites, Beijing University of Chemical Technology, Beijing 100029, PR China. [5]These authors contributed equally: Xiaoyu Tian, Renjie Ren. ✉e-mail: lzhuang@whu.edu.cn; wsheng@tongji.edu.cn

Ni-based catalysts usually dramatically drops at ~0.1 V versus the reversible hydrogen electrode (RHE) due to the Ni surface passivation[12,15], while study shows that HOR catalysts should be able to remain stable up to at least 0.3 V for a substantial power density under practical operating conditions[16]. More severely, the anode potential would be driven up to ~0.8–0.9 V during the start-up and shut-down (SUSD) cycles or upon the $H_2$ starvation events during fuel cell operations[17]. Recently, the amorphous $Ni_{52}Mo_{13}Nb_{35}$ metallic glass was reported with the deactivation potential (the potential at which the HOR current starts to decrease) substantially increased to 0.8 V. Yet, the AEMFC using the $Ni_{52}Mo_{13}Nb_{35}$ anode still shows a significant output cell voltage decay (~38%, from 0.74 to 0.46 V at 200 mA cm$^{-2}$) in 50 h[18]. So far, none of the current Ni-based electrocatalysts can survive the harsh anodic conditions. Therefore, with achieved acceptable activity, efforts need to be geared to improve the stability of non-precious electrocatalysts.

The deactivation of HOR on Ni-based electrocatalysts is accompanied with Ni surface oxidation[19–21], which passivates the catalyst surface and limits the electrochemically active window. Tuning the electronic structure of Ni-based materials for a weaker binding strength of Ni towards O/OH is therefore the most applied strategy. $Ni_{5.2}WCu_{2.2}$ ternary alloy[22] and phase-separated Mo-Ni alloy (PS-MoNi)[23] have exhibited elevated deactivation potential to ~0.3 V. Covering a protective layer outside Ni nanoparticles such as a few-layer hexagonal boron nitride (h-BN) could also prevent Ni from oxidation, which was speculated to originate from the weakened binding affinity towards oxygen species[24]. However, these negligible improvements in stability are not sufficient for practical AEMFC operation, and the stability issue is still the key problem to solve.

The interaction between the metal catalyst and the support also plays a vital role in tuning the reactivity of metal catalysts. Study shows that the interfacial charge transfer from $Ni_3N$ to the carbon support can move the deactivation potential from 0.16 V for $Ni_3N$ to 0.26 V for $Ni_3N/C$[25]. Great improvements have been made on Pt group metals using metal oxide as the support. $TiO_2$ partially encapsulated Pt shows extraordinary HOR activity above 1.0 V, wherein Pt alone will be oxidized and lose its HOR activity[26]. Using $TiO_2$ to support Ru (Ru@$TiO_2$) moves the deactivation potential from 0.2 V (Ru) to 0.9 V (Ru@$TiO_2$)[27]. The interaction between Pt/Ru and $TiO_2$, known as the metal-support interaction (MSI), enables an efficient charge transfer from $TiO_2$ to metals as $TiO_2$ is intrinsically an electron-rich semiconductor (negative semiconductor)[28]. Correspondingly, it is reasonable to speculate that the charge transfer introduced by the MSI may also exist between Ni-based non-precious metals and $TiO_2$. Here, we report that the MSI between $Ni_4Mo$ and the $TiO_2$ support boosts the durability for HOR up to 1.2 V, enabling a continuous AEMFC power output at 400 mA cm$^{-2}$ for nearly 100 h, which makes a step forward for the automotive applications of the AEMFCs.

## Results

### HOR catalytic performance of $Ni_4Mo/TiO_2$

We prepared $Ni_4Mo/TiO_2$ catalysts through annealing the pre-mixed NiMo hydroxide precursor and the $TiO_2$ support in $H_2$ at 400 °C (see Methods). Optimization was made by adjusting the molar ratio between $TiO_2$ and $Ni_4Mo$ according to their HOR performance (Supplementary Fig. S1 and Supplementary Table S1). The HOR catalytic performance of $Ni_4Mo$ and $Ni_4Mo/TiO_2$ catalysts was evaluated using the rotating disk electrode (RDE) method. Figure 1a shows the positive-going sweeps of the cyclic voltammograms (CVs) of $Ni_4Mo$ and the best performing $Ni_4Mo/TiO_2$ in both $H_2$ and $N_2$-saturated 0.1 M NaOH. $Ni_4Mo$ reaches the limiting current density of 2.65 mA cm$^{-2}_{geo}$ at only 85 mV overpotential, showing a very high HOR activity. Yet, the HOR current quickly drops at 0.2 V, and eventually tracks the CV curve collected in $N_2$, which strongly suggests that the surface oxidation of $Ni_4Mo$ blocks the active surface areas and prohibits the hydrogen

oxidation. In contrast, while $Ni_4Mo/TiO_2$ shows a similar HOR limiting current density of 2.23 mA cm$^{-2}_{geo}$ at 90 mV overpotential, it can catalyze the HOR even up to 1.0 V, with a mere 10% limiting current density decay. There is continuous HOR current at even ~1.4 V before oxygen starts to evolve (Supplementary Fig. S2). The difference in the polarization curves in $H_2$ and $N_2$-saturated electrolytes confirms that the anodic current in the presence of $H_2$ in the full potential window is indeed originated from the $H_2$ oxidation. It is also clearly seen from the CV curves collected in $N_2$-saturated electrolyte that $Ni_4Mo/TiO_2$ hardly exhibits any features associated with the surface oxidation as $Ni_4Mo$. Further chronoamperometry measurements verify that $Ni_4Mo/TiO_2$ exhibits a stable HOR current at as high as 1.2 V for 8000 s without noticeable decay (Fig. 1b and Supplementary Fig. S3). In comparison, $Ni_4Mo$ loses 87% of the HOR current when changing the holding potential from 0.2 to 0.3 V, and demonstrates an unacceptable low activity at 0.3 V (Fig. 1b).

Supplementary Fig. S4 shows the HOR polarization curves of $Ni_4Mo$ and $Ni_4Mo/TiO_2$ at different rotation speeds. Koutecky–Levich plots at 0.05 V exhibit a linear relationship between the inverses of $i$ and $\omega^{1/2}$, with the slopes being 5.38 and 5.15 cm$^2$ mA$^{-1}$ s$^{-1/2}$ for $Ni_4Mo$ and $Ni_4Mo/TiO_2$ (insets of Supplementary Fig. S4c, d). These values match reasonably well with the theoretical value of 4.87 cm$^2$ mA$^{-1}$ s$^{-1/2}$ for the 2 e$^-$ HOR[5], and are also in close agreement with the previous study[29]. The exchange current density ($i_0$) was extracted by fitting the kinetic current to the Butler–Volmer equation (Supplementary Fig. S5). The mass activity ($i_{0,m,298 K}$) was then obtained by normalizing $i_0$ to the Ni mass, as shown in Supplementary Table S1 and Supplementary Fig. S6. $Ni_4Mo$ has a $i_{0,m,298 K}$ of $9.6 \pm 0.5$ A g$^{-1}_{Ni}$, in reasonably good agreement with previously reported values (6.8 A g$^{-1}_{Ni}$ in ref. 13 and 14.1 A g$^{-1}_{Ni}$ in ref. 14). The $i_{0,m,298 K}$ of $Ni_4Mo/TiO_2$ first increases, and then decreases with increasing Ti/Ni ratio. At low Ti/Ni ratios (Ti/Ni < 0.4), $Ni_4Mo/TiO_2$ follows similar HOR behavior as $Ni_4Mo$ that it starts to deactivate at ~0.2 V (Supplementary Fig. S1), despite the slightly higher mass activities. The best performance of $Ni_4Mo/TiO_2$ is achieved at Ti/Ni = 0.42, wherein $Ni_4Mo/TiO_2$ exhibits a similar $i_{0,m,298 K}$ (10.1 ± 0.9 A g$^{-1}_{Ni}$) as $Ni_4Mo$, but a more stable HOR current above 1.0 V. When Ti/Ni ratio continues to increase (Ti/Ni > 0.5), in spite of its improved stability, $Ni_4Mo/TiO_2$ loses its mass activity significantly, which is most likely due to the lost active surface area by the $TiO_2$ coverage. The HOR activation energies ($E_a$) on $Ni_4Mo$ and $Ni_4Mo/TiO_2$, determined from the Arrhenius plots (Supplementary Fig. S7), are 15.9 kJ mol$^{-1}$ and 19.5 kJ mol$^{-1}$ respectively, matching well with 18.6 kJ mol$^{-1}$ for $Ni_4Mo$ reported previously[13]. Both $Ni_4Mo$ and $Ni_4Mo/TiO_2$ demonstrate decreased $E_a$ with respect to metallic Ni (30.0 kJ mol$^{-1}$) and partially oxidized Ni (26.0 kJ mol$^{-1}$)[30].

Figure 1c and Supplementary Table S2 summarize recent important progress in the development of Ni-based non-precious metal electrocatalysts for the alkaline HOR. Despite previous efforts in improving the HOR activity by one order of magnitude, achievements in enhancing the deactivation potential remains very limited. With the acceptable high mass activity, the $Ni_4Mo/TiO_2$ reaches a new deactivation potential of 1.2 V for the alkaline HOR Fig. 1.

### AEMFC performance of the $Ni_4Mo/TiO_2$ anode catalyst

Encouraged by the excellent intrinsic HOR performance of the $Ni_4Mo/TiO_2$ catalyst in the RDE measurements, the assembled single cell test was conducted for AEMFC performance. The $Ni_4Mo$ and $Ni_4Mo/TiO_2$ were used as the anode catalysts, the commercial Pt/C was employed as the cathode catalyst, and QAPPT was applied as both anion exchange membrane and ionomer to fabricate the membrane electrode assembly (MEA). Figure 2a shows the cell voltage and power density of the cells using $Ni_4Mo$ and $Ni_4Mo/TiO_2$ as the anode catalysts. $Ni_4Mo$ approaches a peak power density (PPD) of 188 mW cm$^{-2}$, which is prominent in Ni-based non-precious metals and is much higher than the NiMo/KB reported previously (120 mW cm$^{-2}$)[15]. Strikingly, $Ni_4Mo/$

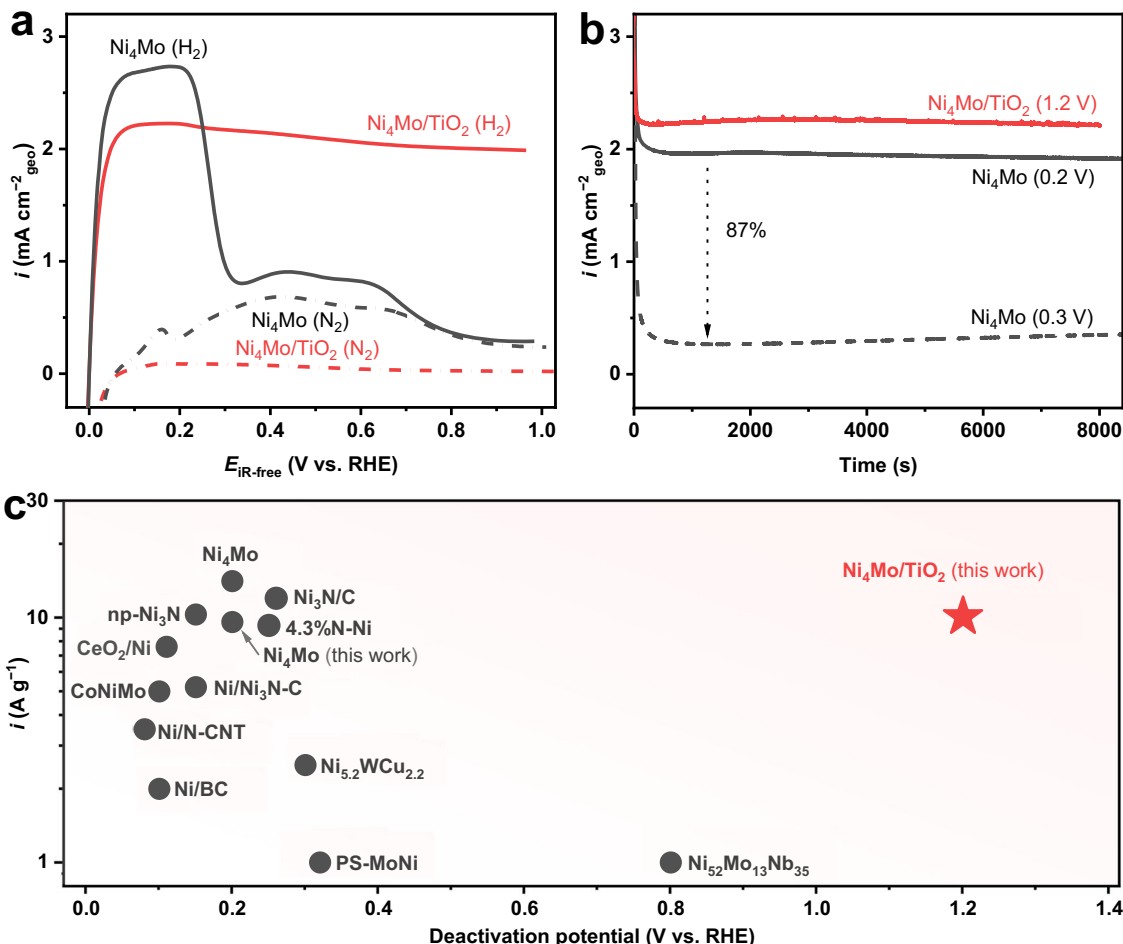

**Fig. 1 | HOR performance of Ni₄Mo/TiO₂. a** Positive-going sweeps of the cyclic voltammograms of Ni₄Mo and Ni₄Mo/TiO₂ recorded in H₂ and N₂-saturated 0.1 M NaOH at 1600 r.p.m. with a scanning rate of 0.5 mV s⁻¹. The potentials are *iR*-corrected; **b** chronoamperometry curves of Ni₄Mo and Ni₄Mo/TiO₂ at constant potentials in H₂-saturated 0.1 M NaOH at 1600 r.p.m. The potentials are not *iR*-corrected; and **c** deactivation potentials and mass activities of Ni-based non-precious metal electrocatalysts for the alkaline HOR. Details are listed in Supplementary Table S2. Note 1: The Ni loadings are 477 and 376 μg_Ni cm⁻²_geo for Ni₄Mo and Ni₄Mo/TiO₂. Note 2: The data of CoNiMo[12] and 4.3%N-Ni[79] were calculated based on the original data. Note 3: The mass activities of bulky PS-MoNi[23] and Ni₅₂Mo₁₃Nb₃₅[18] are unavailable. Here we set them at 1 A g⁻¹ for comparison.

TiO₂ boosts the PPD to 520 mW cm⁻², among the best AEMFC performances that have been reported under similar conditions (see Supplementary Table S3 for a summary of AEMFC performance of non-precious metal anode catalysts). In addition, Ni₄Mo can only approach a current density of 275 mA cm⁻² (at 0.685 V), and fails to operate beyond this point because of the Ni oxidation induced deactivation. However, Ni₄Mo/TiO₂ is able to deliver a much higher current density of 900 mA cm⁻² (at 0.543 V), indicating that Ni₄Mo/TiO₂ is more resistant to oxidation and remains active at a higher polarization potential. The long-term durability of Ni₄Mo/TiO₂ was evaluated at a large current density of 400 mA cm⁻² (Fig. 2b), and the cell exhibits a stable operation for nearly 100 h. Gao et al. have demonstrated that the Ni@CNₓ anode AEMFC exhibits a stable performance at 200 mA cm⁻² for 100 h, which has been considered as a groundbreaking achievement at a large current density in AEMFCs[31]. The Ni₄Mo/TiO₂ anode further elevates the durability to a larger current density of 400 mA cm⁻² for nearly 100 h (also see Supplementary Table S3 for Ni-based anode AEMFC durability). Moreover, the Ni₄Mo/TiO₂ anode can operate at 0.65 V (typical fuel cell operating potential for the automotive applications) for more than 80 h (Fig. 2b), paving the way for progress in the automotive applications of AEMFCs. Furthermore, the cells using Ni₄Mo and Ni₄Mo/TiO₂ as the anode catalysts were discharged consecutively for three times. While the PPD of Ni₄Mo is severely degraded from 188 mW cm⁻² (1st cycle) to 171 mW cm⁻² (2nd

cycle) and 77 mW cm⁻² (3rd cycle), as shown in Fig. 2c, Ni₄Mo/TiO₂ demonstrates a mere decrease of 7% in PPD from 520 mW cm⁻² (1st cycle) to 480 mW cm⁻² (2nd cycle), and retains the performance in the following cycle (Fig. 2d).

## Crystallographic and Morphologic Characteristics of Ni₄Mo/TiO₂

The X-ray diffraction (XRD) pattern in Fig. 3a shows that the synthesized Ni and Mo alloy exhibits a Ni₄Mo phase structure (JCPDS 65-5480), consistent with the ICP-OES results (Supplementary Table S1). The TiO₂ distinctly presents both the anatase phase (JCPDS 89-4921) and the rutile phase (JCPDS 89-0552). The high intensity diffraction peaks at 25.3° and 37.8°, indexed to the anatase TiO₂ (101) and (004) crystal planes, suggest that the anatase phase is the major composition in the pristine TiO₂. Ni₄Mo/TiO₂ exhibits crystallographic features associated with both Ni₄Mo and TiO₂. Increasing the Ti/Ni ratio leads to the decreased relative peak intensity of Ni₄Mo to TiO₂ (Supplementary Fig. S8), corresponding to the gradually decreased Ni₄Mo loading on TiO₂. The transmission electron microscopy (TEM) image shows an interconnected particle morphology of Ni₄Mo with the particle size of about 7.5 nm (Fig. 3b and Supplementary Fig. S9). Aberration-corrected high-angle annular dark-field scanning transmission electron microscope (HAADF-STEM) image of Ni₄Mo (Fig. 3c) shows the 0.210 and 0.202 nm-lattice fringes, corresponding to the (121) and

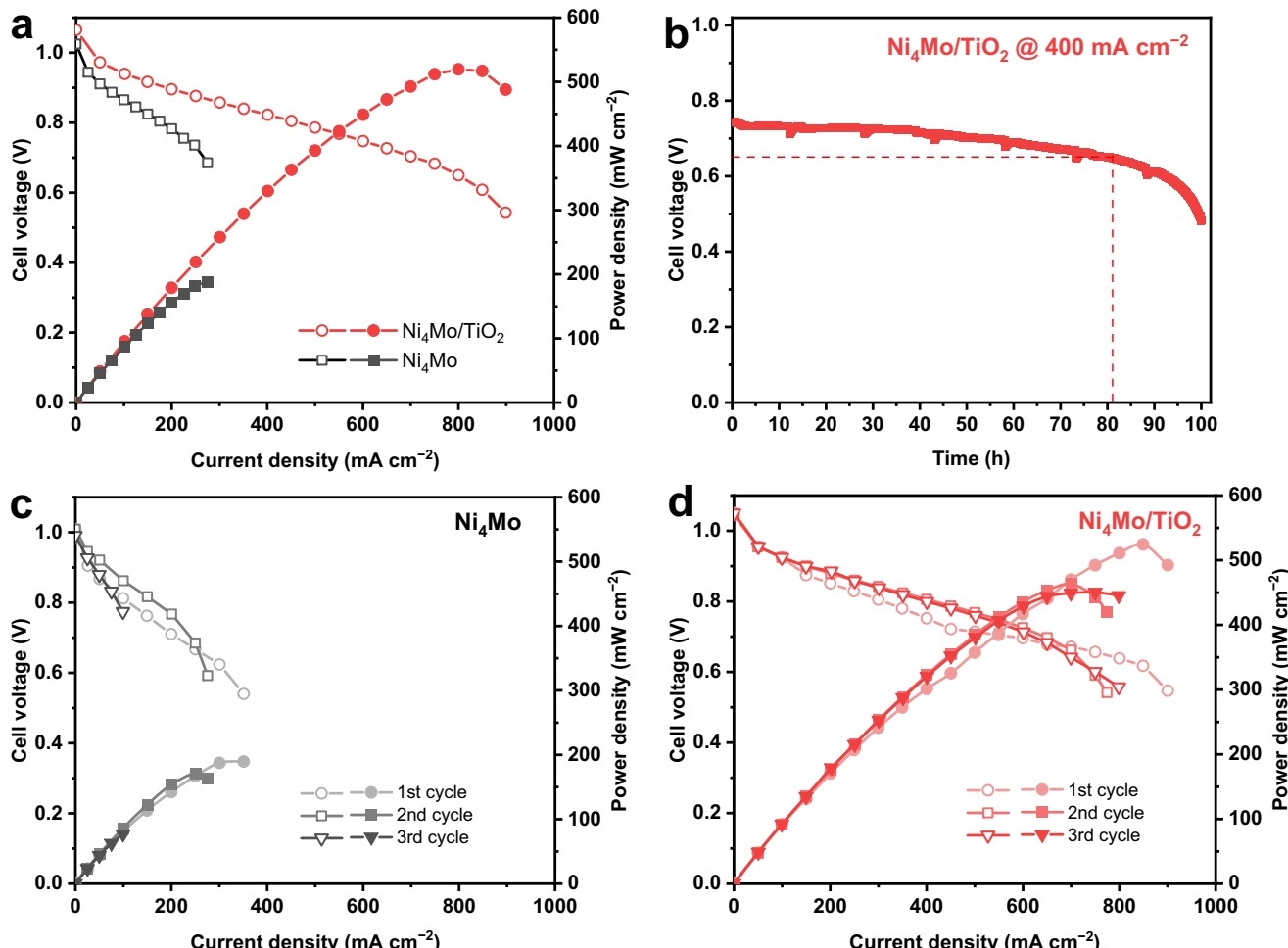

**Fig. 2 | H$_2$-O$_2$ AEMFC performance. a** Polarization and power density curves; **b** AEMFC durability of Ni$_4$Mo/TiO$_2$ at 400 mA cm$^{-2}$; and **c, d** polarization and power density curves of Ni$_4$Mo and Ni$_4$Mo/TiO$_2$ AEMFCs discharged for three times. AEMFC performance test conditions (**a, c** and **d**): cell temperature at 80 °C under H$_2$ and O$_2$ condition with a backpressure of 0.2 MPa for the anode and cathode, H$_2$ and O$_2$ humidified at 80 °C (100% RH) supplied to the anode and cathode compartments with a flow rate of 1000 sccm. AEMFC durability test conditions (**b**): H$_2$ and O$_2$ flow rate of 300 sccm and 500 sccm respectively, under otherwise identical conditions. The cell voltages were recorded with no *iR*-correction. The Ni loadings are 1.35 mg$_{Ni}$ cm$^{-2}$ for Ni$_4$Mo and Ni$_4$Mo/TiO$_2$ at the anode, and the Pt loading is 0.4 mg$_{Pt}$ cm$^{-2}$ for Pt/C at the cathode.

(220) planes of Ni$_4$Mo. The corresponding fast Fourier transform (FFT) pattern demonstrate the tetragonal Ni$_4$Mo crystalline phase with Ni$_4$Mo (121), (220) and (330) planes (Fig. 3d). Elemental mappings using energy dispersive X-ray spectroscopy (EDS) further reveal the compositional distributions of Ni and Mo elements, confirming the formation of a uniform alloy (Supplementary Fig. S10). The pristine TiO$_2$ has a rectangular plate morphology with the 0.354 and 0.237 nm-lattice fringes corresponding to the (101) and (004) planes of the anatase phase (Supplementary Fig. S11). TEM and HAADF-STEM images of Ni$_4$Mo/TiO$_2$ show a uniform distribution of spherical Ni$_4$Mo particles with a mean size of 7.6 nm on the TiO$_2$ support (Fig. 3e–g and Supplementary Fig. S12), maintaining the crystallographic characteristics as unsupported Ni$_4$Mo with the lattice fringe interplanar spacings of 0.210 and 0.202 nm for Ni$_4$Mo (121) and (220) planes. Besides, the lattice fringe with a spacing of 0.237 nm for the TiO$_2$ support is consistent with the pristine TiO$_2$, corresponding to the (004) plane. This structure was further identified by the FFT images (Fig. 3h, i). The inverse FFT image in Fig. 3j also show the interface between TiO$_2$ and Ni$_4$Mo, further demonstrating that Ni$_4$Mo is supported on the TiO$_2$.

## MSI in Ni$_4$Mo/TiO$_2$

Raman spectroscopy study of TiO$_2$ shows six Raman-active modes at around 140 cm$^{-1}$ ($E_g$), 195 cm$^{-1}$ ($E_g$), 393 cm$^{-1}$ ($B_{1g}$), 511 cm$^{-1}$ ($A_{1g} + B_{1g}$) and 635 cm$^{-1}$ ($E_g$) (Supplementary Fig. S13a), in good agreement with Raman spectrum of the anatase TiO$_2$[32]. No detectable Raman shift or peak broadening is observed for TiO$_2$ treated at 400 °C in H$_2$ (TiO$_2$-H$_2$-400 in Supplementary Fig. S13b), indicating that there are not enough oxygen vacancies being introduced, as they will cause the blue-shift and peak broadening in the Raman spectra of TiO$_2$[33,34]. When Ni$_4$Mo is decorated on the TiO$_2$ support, there appears to be distinct blue-shift in the vibrational mode of $E_g$ from 140 cm$^{-1}$ (TiO$_2$) to 150 cm$^{-1}$ (Ni$_4$Mo/TiO$_2$) and peak broadening (Supplementary Fig. S13b), similar to that for Au or Ag decorated TiO$_2$ in a previous study[35]. The blue-shift and peak broadening are most likely due to a compressive strain[36,37], possibly introduced by lattice mismatch between the metal particles and the TiO$_2$ support. The Raman results, together with the aforementioned electron microscopy study, confirm an interfacial interaction between the Ni$_4$Mo particles and the TiO$_2$ support.

Figure 4a shows that X-ray photoemission spectrum (XPS) of the Ni 2$p_{3/2}$ level of Ni$_4$Mo contains four peaks, which can be assigned to Ni$^0$ (852.7 eV), Ni$^{2+}$ (856.1 eV) and their two satellites (858.9 eV for Ni$^0$ satellite and 861.6 eV for Ni$^{2+}$ satellite)[38–40]. Notably, both of the Ni$^0$ and Ni$^{2+}$ peaks of Ni$_4$Mo/TiO$_2$ at all Ti/Ni ratios shift negatively to lower binding energies (BE) (Fig. 4a and Supplementary Fig. S14a). XPS spectra of Mo 3$d$ level reveal a complex Mo 3$d_{3/2}$ and Mo 3$d_{5/2}$ doublets, with the 3$d_{5/2}$ peaks centered at 228.0, 229.0, 230.0 and 232.2 eV,

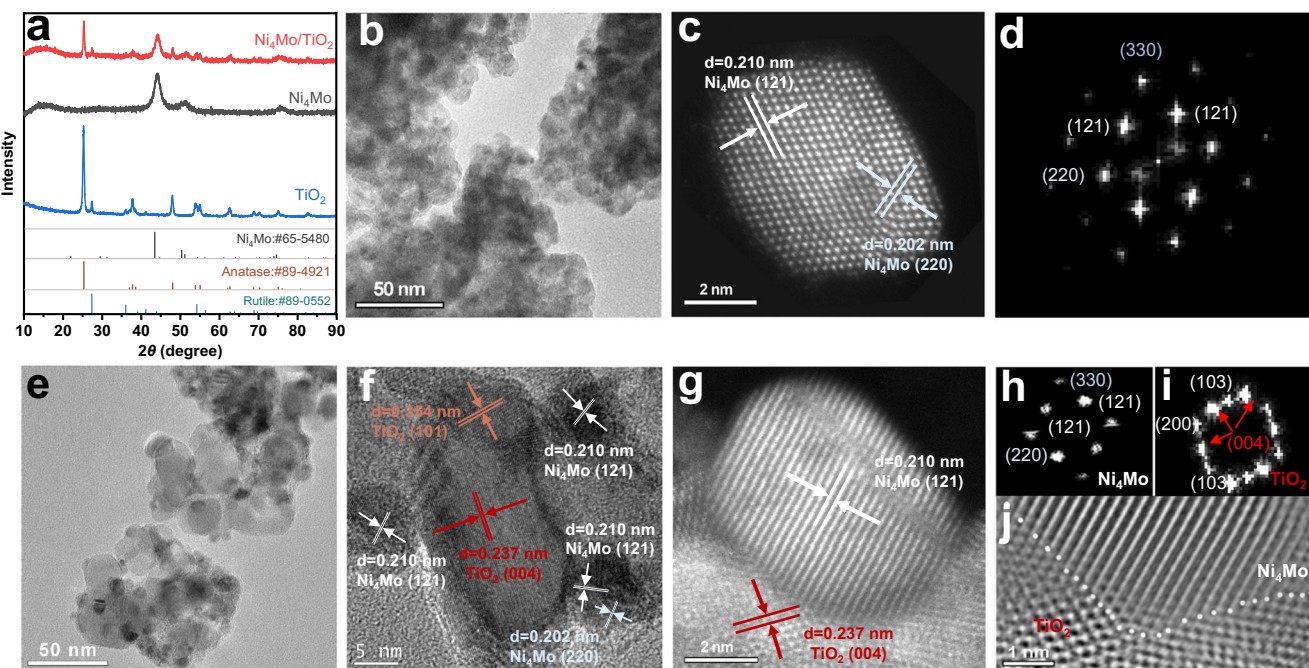

**Fig. 3 | Physical characterizations. a** XRD patterns of $Ni_4Mo$, $Ni_4Mo/TiO_2$ and $TiO_2$; **b** TEM image, **c** HAADF-STEM image, and **d** corresponding fast Fourier transform (FFT) pattern of $Ni_4Mo$; **e** TEM image, **f** HR-TEM image, **g** HAADF-STEM image, **h** and **i** corresponding FFT patterns, and **j** inverse FFT pattern of $Ni_4Mo/TiO_2$.

corresponding to $Mo^0$, $Mo^{4+}$, $Mo^{5+}$ and $Mo^{6+}$ respectively (Supplementary Fig. S14b)[41–43]. These peaks do not shift when loading $Ni_4Mo$ onto $TiO_2$ at all Ti/Ni ratios. Ti 2p doublets of $TiO_2$ show two peaks centered at 458.6 and 464.3 eV, assigned to $Ti^{4+}$ $2p_{3/2}$ and $Ti^{4+}$ $2p_{1/2}$ respectively[44], and the $Ti^{4+}$ in $Ni_4Mo/TiO_2$ displays the positive core level shift to 458.9 eV ($Ti^{4+}$ $2p_{3/2}$) and 464.6 eV ($Ti^{4+}$ $2p_{1/2}$), indicating that Ti is in an electron-deficient status with a higher valence state (Fig. 4b). The O 1s spectrum for $TiO_2$ is de-convoluted to two peaks at 529.9 and 531.2 eV (Fig. 4c), associated with the lattice oxygen in $TiO_2$ (Ti-O) and the surface hydroxyl group (H-O)[27,44]. The O 1s spectrum of Ti-O for $Ni_4Mo/TiO_2$ also exhibits a positive BE shift by ~0.2 eV. Note that $Ni_4Mo/TiO_2$ experienced high temperature reduction in $H_2$ at 400 °C; both of the Ti 2p and O 1s spectra of the bare $TiO_2$ treated under the same condition ($TiO_2$-$H_2$-400) were also collected, and no significant difference was observed (Supplementary Fig. S15). At all Ti/Ni ratios, both of the Ti 2p and O 1s peaks shift to higher BE. The X-ray absorption fine structure spectroscopy (XAFS) is used to further probe the impact of the $TiO_2$ support on the chemical environment of Ni. Notably, the absorption edge of $Ni_4Mo/TiO_2$ displays a slight shift toward the lower photon energy relative to $Ni_4Mo$ (Fig. 4d, left inset), indicating the electron enrichment on Ni atoms in $Ni_4Mo/TiO_2$. The white line absorption intensity is also weaker than that of $Ni_4Mo$ (Fig. 4d, right inset), signifying the lower Ni valence state in $Ni_4Mo/TiO_2$ (Supplementary Fig. S16). In addition, as shown in the Fourier-transform of Ni K-edge extended X-ray absorption fine structure (EXAFS) (Fig. 4e), the intensity of the peak at 2.0 Å, assigned to Ni-Ni/Ni-Mo coordination of $Ni_4Mo/TiO_2$, is higher than that of $Ni_4Mo$, demonstrating an increased Ni coordination number, which is speculated to originate from the interaction with the $TiO_2$ support. XPS and XAFS results strongly suggest that there exists electronic interaction between $Ni_4Mo$ and the $TiO_2$ support through the charge transfer from $TiO_2$ to Ni.

Work function values determined from ultra-violet photoemission spectroscopy (UPS) measurements also support the charge transfer from $TiO_2$ (3.6 eV) to $Ni_4Mo$ (4.3 eV) (Supplementary Fig. S17). To collectively elucidate the electronic interaction between $Ni_4Mo$ and $TiO_2$, the d band density of states was further investigated by UPS as shown in Fig. 4f and Supplementary Fig. S18. All samples illustrate intense emission between −3 and −10 eV below the Fermi level ($\varepsilon_F$), related to O 2p orbitals[44,45]. $TiO_2$ and $TiO_2$-$H_2$-400 do not show density of states at the $\varepsilon_F$, but a very weak emission at −0.7 to −0.9 eV, which probably originates from the $Ti^{3+}$ defect states[44]. The bands located at 0 to −3 eV are attributed to the Ni 3d states[46,47]. The Ni 3d band centers are roughly estimated to be −1.31 eV and −1.59 eV for $Ni_4Mo$ and $Ni_4Mo/TiO_2$ respectively, according to Eq. (1) (see Methods)[48–50]. Clearly, the centroid d band of $Ni_4Mo/TiO_2$ is regulated by $TiO_2$, and down shifts away from the $\varepsilon_F$ compared to $Ni_4Mo$. Considering that the UPS measurement only probes the occupied d states, we later performed theoretical density functional theory (DFT) calculations to identify the density of states (DOS) including the unoccupied states in the next section.

## Robust structure of $Ni_4Mo/TiO_2$ in electrochemical measurements

Post-reaction characterizations were performed to study the structural stability of the $Ni_4Mo/TiO_2$ catalyst. After long-term stability test on the RDE, $Ni_4Mo/TiO_2$ still shows an interconnected particle morphology (TEM and HR-TEM in Supplementary Figs. S19a, b) with the $Ni_4Mo$ phase structure well maintained (selected-area electron diffraction, SAED in Supplementary Fig. S19c), and the Ni, Mo, Ti and O elements also have a uniform spatial distribution (EDS mappings in Supplementary Fig. S19d–h). XPS measurements similarly demonstrate inconspicuous change in the Ni $2p_{3/2}$, Mo 3d, Ti 2p and O 1s spectra after long-term stability test (Supplementary Fig. S20). ICP-MS results suggest that Ni and Ti dissolutions are negligible in 0.1 M NaOH. Yet, Mo is found to dissolve from the top surface of $Ni_4Mo$ particles (Supplementary Table S4), in good agreement with previously reported Mo dissolution on NiMo alloys[51–54]. The $TiO_2$ support does not exhibit an inhibitive effect on Mo dissolution. However, Mo dissolution does not seem to deteriorate the HOR performance, as evidenced by the stable HOR current in the long term CA test (Fig. 1b).

After 100 h durability test in the AEMFC setup, the crystal, morphological, compositional, and electronic structures of $Ni_4Mo/TiO_2$ also exhibit negligible change (XRD, TEM, HR-TEM, SAED, EDS and XPS

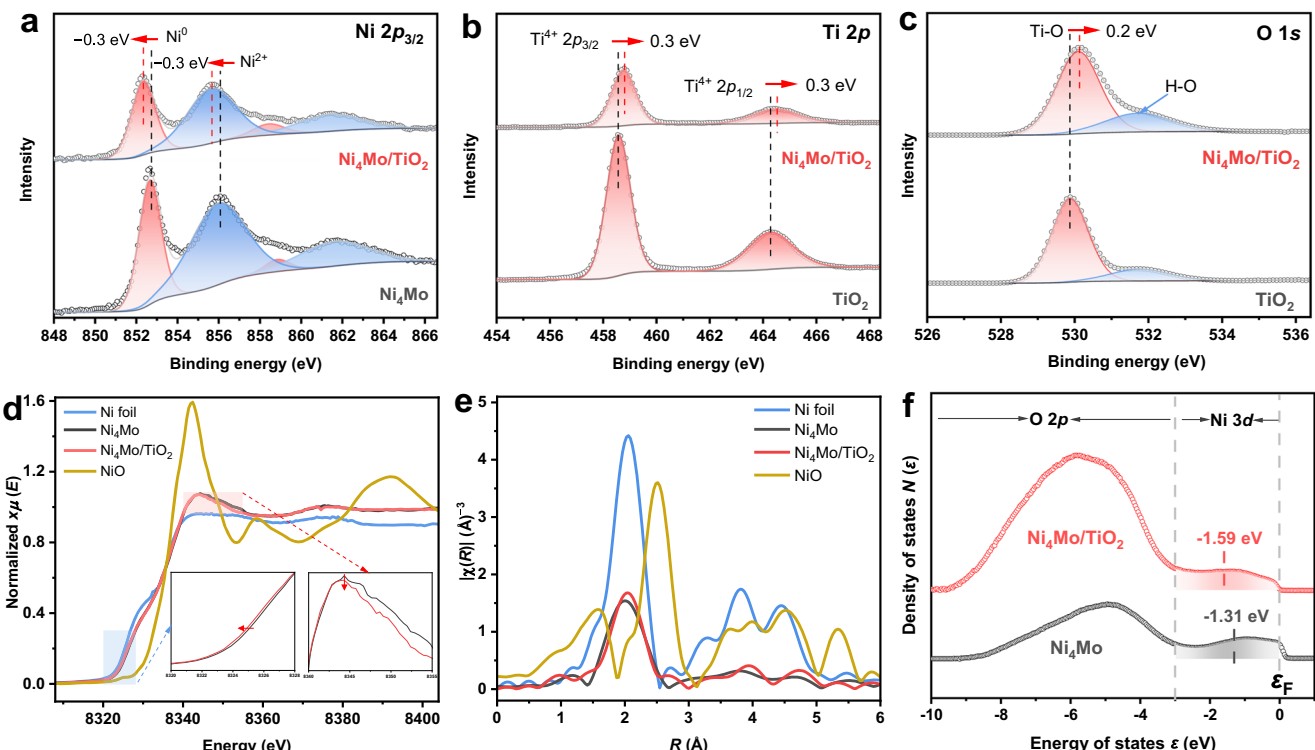

**Fig. 4 | Metal-support interaction in Ni₄Mo/TiO₂. a** Ni $2p_{3/2}$, **b** Ti $2p$ and **c** O $1s$ level X-ray photoemission spectra; **d** normalized Ni K-edge X-ray absorption spectra (the left inset is the magnified near-edge, and the right inset is the white line); **e** Fourier-transform of Ni K-edge EXAFS spectra; and **f** ultra-violet photoemission spectra of Ni₄Mo and Ni₄Mo/TiO₂.

in Supplementary Figs. S21–23). The multiple post-reaction characterizations clearly demonstrate the structural robustness of Ni₄Mo/TiO₂, which enables its high HOR performance and durability in both RDE and AEMFC tests.

**Proposed mechanism for the enhanced stability of Ni₄Mo/TiO₂**
The above results strongly suggest that the MSI between Ni₄Mo and the TiO₂ support modulate the electronic structure of Ni₄Mo, causing a down-shifted $d$ band center, which would weaken the adsorption strength of simple intermediate/molecule on Ni₄Mo, such as H, O and CO, according to the $d$ band theory[55]. Parallel hydrogen-temperature-programmed desorption (H₂-TPD) results clearly demonstrate a lower desorption temperature on Ni₄Mo/TiO₂ than that on Ni₄Mo (Supplementary Fig. S24), signifying a weakened hydrogen binding strength on Ni₄Mo/TiO₂. In line with the reduced hydrogen binding strength, although it is difficult to obtain the specific activity due to the uncertainty in the electrochemical surface area (ECSA) determination, it is reasonable to argue that Ni₄Mo/TiO₂ would exhibit a higher specific activity than Ni₄Mo, as Ni₄Mo/TiO₂ has a similar Ni₄Mo particle size and mass activity, but smaller ECSA due to the coverage of TiO₂. Remarkably, oxygen-temperature-programmed oxidation (O₂-TPO) results exhibit a much higher oxidation temperature of Ni₄Mo/TiO₂ than Ni₄Mo (Supplementary Fig. S25), indicating a stronger oxidation resistance of Ni₄Mo/TiO₂, also in agreement with the hypothesis.

Quasi in situ electrochemical XPS and in situ electrochemical Raman experiments were performed to explore the electronic structures and surface conditions of the Ni₄Mo and Ni₄Mo/TiO₂ catalysts during the HOR process. After potential cycling between −0.1 to 0.2 V for a few cycles to active the surface, the Ni $2p_{3/2}$ and Ti $2p$ level XPS spectra were taken at the open circuit potential (OCP) and oxidizing potentials up to 1.2 V (Supplementary Figs. S26 and S27). The Ni $2p_{3/2}$ spectrum (Supplementary Fig. S26a) of Ni₄Mo collected at OCP shifts by −0.5 eV relative to the ex situ experimental data as the Ni atoms are

negatively charged during the surface activation process, and can be described as the electron-enriched Ni metal (Ni$^{δ-}$)[56]. The BE of Ni⁰ (Ni$^{δ-}$) $2p_{3/2}$ in both Ni₄Mo and Ni₄Mo/TiO₂ stays constant at all potentials, while it is negatively shifted by 0.2-0.3 eV in Ni₄Mo/TiO₂ with respect to that in Ni₄Mo regardless of the applied potentials (Fig. 5a and Supplementary Fig. S26a, b). The Ni²⁺ content estimated from XPS data is 21% in Ni₄Mo at the initial OCP, and dramatically increases to 44% at 1.2 V, while the Ni²⁺ content is substantially reduced in Ni₄Mo/TiO₂ (21% at OCP to 33% at 1.2 V, Supplementary Fig. S26c). The correlation between the HOR polarization curves and the Ni⁰ (Ni$^{δ-}$) percentages shows that the severely decreased current on Ni₄Mo at 0.3 V matches relatively well with the decreased Ni⁰ (Ni$^{δ-}$) content (Supplementary Fig. S26d), indicating that Ni₄Mo deactivation is originated from Ni oxidation. The Ti⁴⁺ $2p_{3/2}$ spectra demonstrate a ~0.2 eV positive BE shift from OCP to 1.2 V (Fig. 5b and Supplementary Fig. S27a). Interestingly, a new peak signal in Ti $2p$ spectra of Ni₄Mo/TiO₂ emerges at the BE lower than Ti⁴⁺ $2p_{3/2}$ after the surface activation, which can be assigned to the reductive Ti³⁺ $2p_{3/2}$ species[44]. The Ti³⁺ $2p_{3/2}$ peak shifts to higher BE (456.5 eV at OCP to 457.2 eV at 1.2 V), and decreases in percentage (34% at OCP to 22% at 1.2 V, Supplementary Fig. S27b) with increasing applied potential. Ti³⁺ has been verified to boost the efficient charge transfer from TiO₂ to metals by decreasing the work function of TiO₂ bulk[57,58], and thus may also play an important part in constructing the MSI of Ni₄Mo/TiO₂. The quasi in situ XPS experiments further confirm the existence of the charge transfer from TiO₂ to Ni₄Mo under the electrochemical conditions.

The in situ Raman spectra were also collected after the surface activation (Fig. 5c, d). The bands at 310 and 893 cm⁻¹, which could be respectively assigned to the Mo=O stretching mode and bending mode in the MoO₄²⁻ tetrahedron (see Supplementary Table S5 for the band assignment), exist in the full potential window investigated from the OCP to 1.2 V for both Ni₄Mo and Ni₄Mo/TiO₂, owing to the oxidation of Mo in Ni₄Mo to MoO₄²⁻[52]. The peak at 483 cm⁻¹ is indexed to the

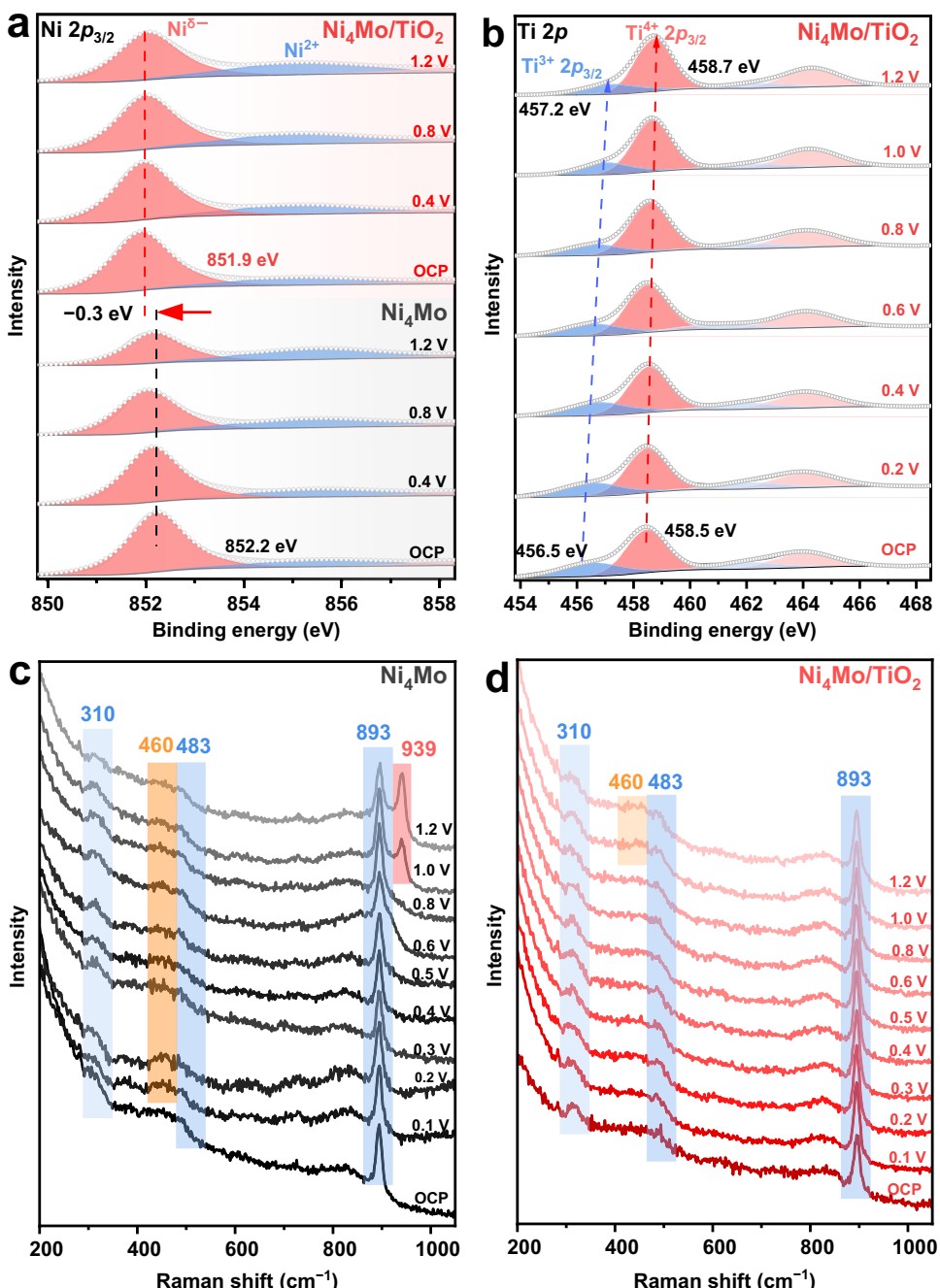

**Fig. 5 | Mechanistic analyses. a**, **b** Quasi in situ electrochemical XPS spectra, and **c**, **d** in situ electrochemical Raman spectra of Ni$_4$Mo and Ni$_4$Mo/TiO$_2$ collected at the selected potentials in 0.1 M NaOH during the HOR. The potentials are *iR*-corrected.

symmetric stretching mode of bridging Mo-O-Mo bond in Mo$_2$O$_7$$^{2-}$, formed from the MoO$_4$$^{2-}$ dimerization[52,59]. Furthermore, the band at 460 cm$^{-1}$ starting from 0.1 V and the one at 939 cm$^{-1}$ starting from 1.0 V on Ni$_4$Mo (Fig. 5c), attributed to the Ni-OH symmetric stretching mode of Ni(OH)$_2$[60–63] and Mo-O stretching mode of NiMoO$_4$[59,64] respectively, indicate the formation of hydroxide and oxide of Ni and Mo during the HOR from 0.1 to 1.2 V. Interestingly, these peaks associated with Ni-OH or Mo-O vibrations disappear on Ni$_4$Mo/TiO$_2$, and only when the potential reaches 1.0 V is the Ni-OH vibration marginally visible (Fig. 5d), revealing that most of the Ni surface atoms likely exist in the form of Ni$^0$ (Ni$^{δ-}$). These experimental results clearly suggest that Ni$_4$Mo/TiO$_2$ has great resistance to oxidation to endure the harsh oxidative conditions at high anodic potentials. This high oxidation resistance is likely due to the strong electronic modulation between

Ni$_4$Mo and TiO$_2$, which renders a down-shifted *d* band center, and in turn weakened OH adsorption energy. DFT calculations also demonstrate a similar trend (Supplementary Fig. S28), in line with the hypothesis.

There have been debates as to the role of the surface hydroxyl species in the alkaline hydrogen oxidation on Pt group metal-based electrocatalysts, wherein the surface OH may (bi-functional mechanism[9]) or may not (HBE mechanism[4,65]) directly participate in the HOR process. However, on Ni-based non-precious materials, it is rather difficult to differentiate the role of the surface OH, as Ni-based materials usually exhibit complicated surface conditions in a broad potential regime, and there is by far no experimental evidence regarding the direct participation of surface OH in the HOR yet. In our opinion, the surface OH most likely acts as the spectator, which

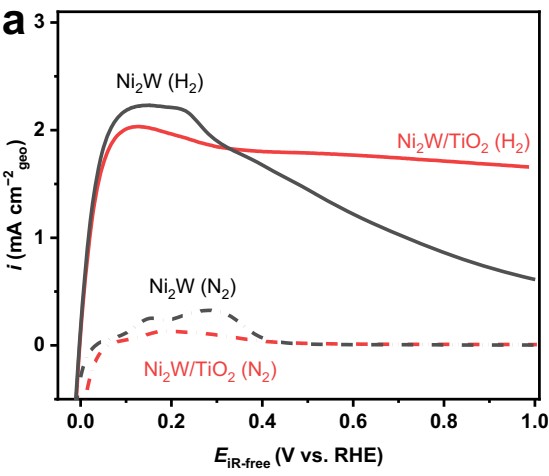
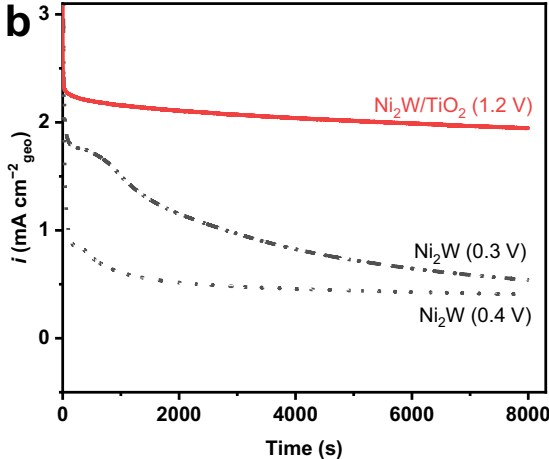

**Fig. 6 | HOR performance of Ni$_2$W/TiO$_2$. a** Positive-going sweeps of the cyclic voltammograms of Ni$_2$W and Ni$_2$W/TiO$_2$ recorded in H$_2$ and N$_2$-saturated 0.1 M NaOH at 1600 r.p.m with a scanning rate of 0.5 mV s$^{-1}$. The potentials are *iR*-corrected; and **b** chronoamperometry curves of Ni$_2$W and Ni$_2$W/TiO$_2$ in H$_2$-saturated 0.1 M NaOH at 1600 r.p.m. The potentials are not *iR*-corrected. The Ni loadings are 349 and 312 $\mu g_{Ni}$ cm$^{-2}_{geo}$ for Ni$_2$W and Ni$_2$W/TiO$_2$.

deteriorates the HOR activity by blocking the active surface area. This speculation is supported by the observations that the HOR current decrease on Ni$_4$Mo and the metal surface oxidization occur concurrently (Fig. 1a and Supplementary Fig. S2), and the appearance of Raman bands associated with Ni(OH)$_2$ and NiMoO$_6$ in the Raman spectra of Ni$_4$Mo. Therefore, it becomes very straightforward to weaken the OH binding strength in designing the HOR catalysts to enhance their anti-oxidation ability. By introducing the MSI using TiO$_2$ as the support, the *d* band center of Ni$_4$Mo is downwards shifted, causing a weakened binding strength to surface O or OH. It is clearly seen from Supplementary Fig. S29, the CV of Ni$_4$Mo/TiO$_2$ is mainly composed of the capacitive current, demonstrating much mitigated surface oxidation. Therefore, we hypothesize that the surface OH plays the role as the blocking species on Ni$_4$Mo, and attribute the improved anti-oxidation ability of Ni$_4$Mo/TiO$_2$ to its much weakened OH adsorption strength.

### CO-tolerance of Ni$_4$Mo/TiO$_2$

The modulated *d* band of Ni$_4$Mo by the TiO$_2$ support not only weakens the OH binding strength, but may also reduce the CO binding energy[55]. As shown in Supplementary Fig. S30, the CO-stripping peak on Ni$_4$Mo/TiO$_2$ moves negatively to 0.50 V in comparison with 0.70 V on Ni$_4$Mo. Moreover, Ni$_4$Mo/TiO$_2$ exhibits HOR activity with a negligible decay in the presence of 2000 p.p.m CO, compared to a -14% decrease in the HOR limiting current density of Ni$_4$Mo (Supplementary Fig. S31a). Chronoamperometry measurements at 0.2 V also demonstrate that the HOR current on Ni$_4$Mo/TiO$_2$ in the presence of 2000 p.p.m CO only decays 14% after 8000 s, much improved than a ~40% decay on Ni$_4$Mo (Supplementary Fig. S31b). The satisfying CO-tolerant capability of Ni$_4$Mo/TiO$_2$ is attributed to its weakened CO binding strength, which originates from the modulated electronic structure of Ni$_4$Mo by the TiO$_2$ support.

### Applicability of the MSI to other Ni-based electrocatalyst

We further investigated whether the MSI between the metal catalyst and the TiO$_2$ support can be applied to other Ni-based electrocatalysts for the enhanced stability. Despite its excellent HOR activity, Ni$_2$W also loses the HOR activity at potentials higher than 0.2 V, in line with the surface oxidation in this potential regime (Fig. 6a). Optimized Ni$_2$W/TiO$_2$ (Ti/Ni = 0.46) not only exhibits good HOR activity at a low overpotential, but maintains the activity at a very small degradation rate (Fig. 6a and Supplementary Fig. S32, also see Methods, Supplementary Table S6, Supplementary Figs. S33–36 for detailed information on the

synthesis and material structure). Although W is also found to slightly leach from the surface of Ni$_2$W nanoparticles in the alkaline electrolyte under the experimental conditions (Supplementary Table S7), long-term stability tests confirm that Ni$_2$W/TiO$_2$ can remain stable activity towards hydrogen oxidation at 1.2 V (Fig. 6b and Supplementary Fig. S37). This finding highlights the strategic importance of the MSI in tailoring the electronic structure of Ni-based electrocatalysts to elevate the HOR stability in alkaline electrolytes.

In conclusion, by applying the MSI between NiM alloys and the TiO$_2$ support, we successfully synthesized Ni$_4$Mo/TiO$_2$ and Ni$_2$W/TiO$_2$ electrocatalysts with enhanced HOR stability. The catalysts not only maintain good HOR mass activities (10.1 ± 0.9 A g$^{-1}_{Ni}$ for Ni$_4$Mo/TiO$_2$ and 6.8 ± 0.2 A g$^{-1}_{Ni}$ for Ni$_2$W/TiO$_2$), but also show stable HOR current at as high as 1.2 V. AEMFC tests, using Ni$_4$Mo/TiO$_2$ as the anode catalyst, demonstrate a good performance with a peak power density of 520 mW cm$^{-2}$ and durability at 400 mA cm$^{-2}$ for nearly 100 h. Detailed structural and electronic analyses confirm the existence of the MSI through charge transfer from the TiO$_2$ support to NiM metals. Modulated electronic structure of NiM weakens the adsorption strength of H, O/OH and CO on NiM surfaces, rendering a high anti-oxidation ability of NiM/TiO$_2$. These results highlight the significance of the MSI in improving the catalyst stability, and evidence the intriguing promise of Ni$_4$Mo/TiO$_2$ as an efficient and robust anode catalyst, making a step forward in the applications of the AEMFC technology.

## Methods

### Material synthesis

The Ni$_4$Mo/TiO$_2$ catalysts were synthesized through a three-step method. Firstly, the NiMo hydroxide precursor was prepared by a solvothermal method[14]. In brief, 735 mg Ni(NO$_3$)$_2$·6H$_2$O (98%, Sinopharm) and 81 mg (NH$_4$)$_6$Mo$_7$O$_{24}$·4H$_2$O (99%, Alfa Aesar) were dissolved in 2.5 mL ultra-pure H$_2$O (18.2 MΩ cm, Millipore), followed by addition of 12.5 mL ethylene glycol (99%, Innochem) and 1 mL NH$_3$·H$_2$O (18% NH$_3$ basis, Sigma-Aldrich). After stirring for 2 h, the solution was transferred into a Teflon-lined stainless steel autoclave and heated at 190 °C for 1 h. When cooling down to the room temperature, the synthesized green powder was washed using ethyl alcohol (99.5%, Innochem) and ultra-pure H$_2$O (1:1 in volume) for five times, and collected by centrifugation. Secondly, six batches of the green precipitate were mixed with 100, 67, 50, 33, 25 and 20 mg TiO$_2$ (Aeroxide® P25, Acros Organic) respectively, and then dispersed in 50 mL ethyl alcohol by magnetically stirring for 20 h. Subsequently, the mixture was collected by vacuum filtration, and grounded in a mortar for 20 min after drying

at 50 °C in vacuum ($2.13 \times 10^4$ Pa). Finally, the collected substance was heated up to 400 °C at 3 °C min$^{-1}$ in a reductive atmosphere ($H_2$/$N_2$ = 1:5 in volume) and remained at 400 °C for 1 h to obtain the $Ni_4Mo$/$TiO_2$ catalysts. $Ni_4Mo$ was prepared using the same method without adding $TiO_2$ in the second step.

Ni$_2$W and Ni$_2$W/TiO$_2$ catalysts were synthesized by the same procedure, except that the initial Ni and W precursors were 735 mg Ni(NO$_3$)$_2$·6H$_2$O (98%, Sinopharm) and 177.5 mg (NH$_4$)$_2$WO$_4$ (99.99%, Alfa Aesar), and the final heat-treatment temperature was 500 °C.

## Physical characterizations

**Electron microscopy.** High-resolution transmission electron microscopy (HR-TEM) measurement was conducted on the FEI Talos 200X and JEOL JEM-2100F transmission electron microscopes at the accelerating voltage of 200 kV. Aberration-corrected high-angle annular dark-field scanning transmission electron microscopy (HAADF-STEM) measurement was carried out on an atomic-resolution analytical microscope (thermalfisher scientific titan themsis Z) equipped with a probe spherical aberration corrector at an acceleration voltage of 300 kV. Energy dispersive X-ray spectroscopy was collected on the Oxford Instruments X-Max 80 T. The sample was made through dispersing the catalysts in ethyl alcohol by sonication. The dispersion was then dropped on an ultrathin carbon grid, and dried in air for few minutes.

**Powder X-ray diffraction (XRD).** XRD measurement was performed on a D8 Advance X-ray diffractometer (Bruker) using Cu Kα radiation ($\lambda$ = 0.15418 nm) at 40 kV and 40 mA. The data were collected with $2\theta$ ranged from 10° to 90° at a scanning rate of 10° min$^{-1}$.

**Raman spectroscopy.** Raman experiment was carried out with a confocal microscope Raman spectrometer (DXR3, Thermo Scientific). The excitation wavelength was 532 nm.

**X-ray photoemission spectroscopy (XPS).** The states of Ni, Mo, Ti and O in the catalysts were examined on a ThermoFischer ESCALAB 250Xi X-ray photoelectron spectrometer equipped with Al X-ray source (Al Kα, 1486.6 eV). All spectra were processed using the Shirley background correction, and calibrated with the C 1$s$ component at 284.8 eV. The Gaussian–Lorentzian line shape was adopted to fit the spectra.

**X-ray absorption fine structure spectroscopy (XAFS).** The XAFS measurements were performed at 1W1B station in Beijing Synchrotron Radiation Facility, operated at 2.5 GeV with a maximum current of 250 mA. The spectra were collected in the fluorescence mode using a Lytle detector. Samples were pelletized with diameter of 8 mm and thickness of 1 mm using the PVDF powder as the binder. The acquired Ni K-edge extended X-ray absorption fine structure (EXAFS) data were processed according to the standard procedures using the Athena and Artemis implemented in the IFEFFIT software packages[66]. The EXAFS spectra were subtracted by the post-edge background from the overall absorption, and normalized with respect to the edge-jump step. The $\chi(k)$ data were Fourier transformed to real ($R$) space using a hanning window (d$k$ = 1.0 Å$^{-1}$) to separate the EXAFS contributions from different coordination shells. Least-squares curve fitting was performed using the Artemis module of IFEFFIT software packages to obtain the quantitative structural parameters. In the fitting, the amplitude reduction factor $S_0^2$ was fixed, and the internal atomic distances $R$, Debye-Waller factor $\sigma^2$, and the edge-energy shift $\Delta E_0$ were allowed to run freely.

**Ultraviolet photoemission spectroscopy (UPS).** Surface electronic structure of the catalysts was investigated on the PHI5000 VersaProbe III electron spectrometer (Scanning ESCA Microprobe) at UV photon

energy of 21.2 eV (He I) under ultra-high vacuum ($4 \times 10^{-6}$ Pa). The total energy resolution was 0.10 eV. Shirley background was subtracted as described in the previous study[48]. The band between −3 eV and −10 eV centered at −4.9 eV for the $Ni_4Mo$ sample was assigned to the O 2$p$ states due to inevitable oxidation upon exposure to air[45], while the band in the same window centered at −6.0 eV for the $Ni_4Mo$/$TiO_2$ sample was O 2$p$ states mainly originated from $TiO_2$[44]. The bands ranged from 0 to −3 eV in both samples were assigned to the Ni 3$d$ photoemission[46]. The $d$ band center energy relative to the Fermi level was calculated based on the following equation[48–50]

$$\varepsilon_{\mathrm{d}} = \int N(\varepsilon)\varepsilon\mathrm{d}\varepsilon \Big/ \int N(\varepsilon)\mathrm{d}\varepsilon \qquad (1)$$

where $N(\varepsilon)$ is the density of states, and $\varepsilon$ is the energy of states.

For the work function calculation, the valence band spectra were also collected at the same spectrometer with a −5 V bias voltage applied to the sample. The work function ($\varphi$) could be calculated from the following equation[67]

$$\varphi = h\nu - E_{\mathrm{cutoff}} \qquad (2)$$

where $h\nu$ is the energy of the UV photon (He I, 21.2 eV), and $E_{\mathrm{cutoff}}$ is the energy of the secondary-electron cutoff.

**H$_2$-temperature programmed desorption (H$_2$-TPD).** Surface property towards H adsorption/desorption of the catalyst was measured using a ChemBET instrument (Quantachrome). The signal was determined by a thermal conductivity detector (TCD). 100 mg catalyst was placed in a quartz tube, and heated from the room temperature up to 400 °C at a ramping rate of 10 °C min$^{-1}$ with flowing He. The sample was kept at 400 °C for 30 min, and then cooled down to the room temperature under He-flow. Subsequently, H$_2$ was introduced into the tube for adsorption until a stable TCD signal. After removing H$_2$ with flowing He, the sample was heated up to 400 °C with a ramping rate of 10 °C min$^{-1}$.

**O$_2$-temperature programmed oxidation (O$_2$-TPO).** Surface property towards binding strength of oxygen-related species was measured on the same instrument as H$_2$-TPD. 100 mg catalyst was placed in a quartz tube, and heated from the room temperature up to 400 °C at a ramping rate of 10 °C min$^{-1}$ with flowing He. The sample was kept at 400 °C for 30 min, and then cooled down to the room temperature under He-flow. Subsequently, the sample was gradually oxidized from the room temperature to 400 °C at a ramping rate of 10 °C min$^{-1}$ in the presence of 3% O$_2$ in He.

**Inductively coupled plasma-optical emission spectrometry (ICP-OES).** The chemical contents of Ni, Mo and Ti were examined by ICP-OES on Agilent-730-OES. In brief, the catalyst was mixed with 5 mL HNO$_3$ (67%, Sinopharm), 1 mL HF (40%, Sinopharm) and 1 mL HCl (37%, Sinopharm) in a Teflon-lined stainless steel autoclave, followed by heating at 180 °C for 8 h. After cooling down to the room temperature, the solution was transferred to a 25-mL volumetric flask, and diluted with ultra-pure H$_2$O to the metered volume for ICP measurement.

**Inductively coupled plasma mass spectrometry (ICP-MS).** ICP-MS (Aglient-7700) was used to detect the Ni, Mo, W and Ti dissolution in the alkaline electrolyte after electrochemical measurements.

## Electrochemical measurement

All electrochemical measurements were performed on the electrochemical workstation (VSP-300, Biologic). The temperature was set at 25 ± 0.5 °C, unless otherwise emphasized. The catalyst ink was prepared by dispersing and sonicating the catalyst in a mixture of 750 μL

isopropanol (99.5%, Innochem), 200 μL ultra-pure $H_2O$ and 50 μL Nafion (5 wt%, Sigma-Aldrich) with a final concentration of 6-8 $mg_{Ni}$ $mL^{-1}$. 10 μL catalyst ink was deposited on a glassy carbon electrode (5 mm in diameter, Tianjin Aida), which was pre-polished with 50 nm alumina slurry (99.0%, Tianjin Aida), and dried in air at the room temperature, resulting in the Ni loading of ~400 $μg_{Ni}$ $cm^{-2}_{geo}$ (see Supplementary Tables S1 and S6 for the specific loadings). The catalyst thin film electrode was then mounted onto a rotator (Pine Instrument), serving as the working electrode. A KCl-saturated calomel electrode (SCE, Tianjin Aida) and a graphite rod (spectral purity, Tianjin Aida) were used as the reference and counter electrodes, respectively. All potentials reported in this paper were referenced to the reversible hydrogen electrode (RHE), which was calibrated by measuring the HER/HOR on a Pt disk (5 mm in diameter, Pine Instrument) in $H_2$-saturated 0.1 M NaOH (99.99% metal trace, Sigma-Aldrich).

**Cyclic voltammetry measurement.** Surface properties of the as-prepared electrocatalysts were investigated using the cyclic voltammetry (CV) method from −0.3 V to 1.8 V versus RHE with a rotating speed of 1600 r.p.m and a scanning rate of 20 mV $s^{-1}$ in $N_2$-saturated 0.1 M NaOH.

**Rotating disk electrode measurement.** HOR performances of the as-prepared electrocatalysts were examined using the RDE technique. The catalyst thin film electrodes were first activated by CV in $H_2$-saturated 0.1 M NaOH between −0.1 and 0.2 V versus RHE at 20 mV $s^{-1}$ for several cycles until a steady polarization curve was obtained. The HOR polarization curves were then collected with a rotating speed of 400, 900, 1600 and 2500 r.p.m and a scanning rate of 0.5 mV $s^{-1}$ to minimize the capacitive charge contribution.

The HOR kinetic current ($i_K$) was calculated based on the Koutecky–Levich equation[68]

$$\frac{1}{i} = \frac{1}{i_K} + \frac{1}{i_D} \tag{3}$$

where $i$ is the measured current, and $i_D$ is the diffusion limited current.

The HOR exchange current ($i_0$) was obtained subsequently by fitting $i_K$ to the Butler–Volmer equation[69]

$$i_K = i_0 \left( e^{\frac{\alpha_a F}{RT}\eta} - e^{\frac{-(1-\alpha_a)F}{RT}\eta} \right) \tag{4}$$

where $i_0$ is the HOR exchange current, $\alpha_a$ is transfer coefficient for the HOR, $F$ is the Faraday constant (96485 C $mol^{-1}$), $R$ is the universal gas constant (8.314 J $mol^{-1}$ $K^{-1}$), $T$ is the temperature, and $\eta$ is the overpotential. $\alpha_a$ was between ~0.1 and ~0.4.

Activation energy ($E_a$) of the HOR was measured at different temperatures from 2 °C to 30 °C. The $E_a$ can be calculated from the Arrhenius equation

$$\log i_0 = \frac{-E_a}{\ln 10 \times R \times T} + const. \tag{5}$$

where $i_0$ (mA $cm^{-2}$) is the exchange current density at different temperatures, $E_a$ is the activation energy (J $mol^{-1}$), $R$ is the universal gas constant (8.314 J $mol^{-1}$ $K^{-1}$), and $T$ is the temperature (K).

**Chronoamperometry measurement.** Chronoamperometry (CA) experiment was taken at a constant potential for 8000 s in $H_2/N_2$-saturated 0.1 M NaOH with a rotating speed of 1600 r.p.m for stability test.

**CO stripping and CO tolerance measurement.** CO stripping was conducted using the CA and CV methods. The potential was held at 0.1 V versus RHE for 30 min, during which time the electrolyte was purged with CO, allowing for complete CO adsorption, followed by 20 min-purging with $N_2$ to remove the remaining CO in the electrolyte. Then CO stripping was performed by taking CV from 0 to 1.2 V versus RHE at a scanning rate of 20 mV $s^{-1}$ for 2 cycles. CO tolerance experiment was performed using the RDE technique and CA method in $H_2$-saturated 0.1 M NaOH at 0.2 V versus RHE in the presence of 2000 p.p.m CO.

**Electrochemical impedance spectroscopy (EIS) measurement.** Solution resistance was obtained after each RDE test by EIS measurement. The EIS spectra were taken at 0 V vs. open circuit potential (OCP) with a 10 mV voltage perturbation, and the frequency was from 100 mHz to 200 kHz. The real part of the resistance at 1 kHz was taken as the solution resistance ($R \approx 40 \, \Omega$).

**In situ electrochemical Raman spectroscopy test**
The in situ electrochemical Raman test was performed on a Renishaw Via Raman microscope with a 532 nm excitation wavelength and a 50× objective. The sample ink was prepared by dispersing the catalyst in a mixture of ultra-pure $H_2O$, isopropanol and Nafion, and then dropped on a piece of carbon paper (CP, 15 × 15 $mm^2$), serving as the working electrode. Prior to collecting the Raman data, the Ni₄Mo and Ni₄Mo/TiO₂ working electrodes were scanned between −0.1 and 0.2 V versus RHE for 10 cycles in $H_2$-saturated 0.1 M NaOH in a home-made Raman cell (Supplementary Fig. S38), equipped with a platinum counter electrode and a SCE reference electrode. The solution resistance was ~26 Ω for the in situ Raman setup. The working electrode was held at each potential (OCP and 0–1.5 V versus RHE at an interval of 0.1 V) for 2 min for the Raman spectra collection. The laser intensity was set to be 10% with a collection time of 50 s for each spectrum.

**Quasi in situ electrochemical X-ray Photoemission Spectroscopy test**
Quasi in situ electrochemical XPS measurement was carried out on the ThermoFischer ESCALAB 250Xi instrument by applying a monochromatic Al Kα X-ray source (1486.8 eV) operating at 12.5 kV and 16 mA under ultra-high vacuum (8 × $10^{-10}$ Pa). The total energy resolution was 0.10 eV. The catalyst ink was prepared by dispersing the catalyst in a mixture of ultra-pure $H_2O$, ethyl alcohol and Nafion, and then dropped on a glassy carbon (GC) electrode (4 mm in diameter), serving as the working electrode. Before collecting the XPS data, the Ni₄Mo and Ni₄Mo/TiO₂ working electrodes were scanned between −0.1 and 0.2 V versus RHE for 10 cycles in $H_2/N_2$ mixture-saturated 0.1 M NaOH in a home-made cell (Supplementary Fig. S39), equipped with a platinum counter electrode and a SCE reference electrode. The solution resistance was ~10 Ω for the quasi in situ XPS setup. The working electrode was first held at each potential (OCP, 0, 0.2, 0.3, 0.4, 0.5, 0.6, 0.8, 1.0 and 1.2 V versus RHE) for 2 min, then vacuumed in the preparation chamber, and finally transferred to the test chamber for XPS spectrum collection without exposure to the air. All spectra were processed using the Shirley background correction, and calibrated with the C 1s component at 284.8 eV. The Gaussian–Lorentzian line shape was adopted to fit the spectra.

**Membrane-electrode assembly and AEMFCs test**
The as-synthesized Ni₄Mo or Ni₄Mo/TiO₂ was used as the anode catalyst and the commercial Pt/C (60 wt%, Johnson-Matthey) was used as the cathode catalyst. The self-designed QAPPT [quaternary ammonia poly (N-methyl-piperidine-co-p-terphenyl)] was applied as both anion exchange membrane and ionomer in the electrodes[70]. For the anode catalyst, the ink was prepared by adding Ni₄Mo or Ni₄Mo/TiO₂, Vulcan XC-72 carbon and 20 mg $mL^{-1}$ ionomer (16 wt%) solution into isopropanol and then sonicated for 30 min. The weight ratio of Ni₄Mo or Ni₄Mo/TiO₂ to Vulcan XC-72 carbon was 4:1. For the cathode catalyst, Pt/C and the 20 wt% ionomer solution was mixed in isopropanol and

then sonicated for 15 min. To make the catalyst-coated membrane (CCM), the catalyst ink was sprayed onto the membrane (25 μm), which was heated at 70 °C to remove the isopropanol. The anode and cathode catalyst loadings were 1.35 $mg_{Ni}$ $cm^{-2}$ and 0.4 $mg_{Pt}$ $cm^{-2}$ respectively, and the catalyst sprayed area was fixed to 4 $cm^2$. In order to exchange the anion of the membrane and ionomer to $OH^-$, the CCM was soaked in 1 M KOH at 60 °C for 24 h, and meanwhile hydrogen was purged into the KOH solution to prevent the catalyst oxidation. Subsequently, the CCM was washed with ultra-pure $H_2O$ for several times to remove the excess KOH. The membrane electrode assembly (MEA) was assembled by placing the CCM between two pieces of carbon paper (AvCarb GDS3250) used as gas diffusion layer without hot-pressing. The AEMFC performance test was conducted on an 850E Multi Range fuel cell test station (Scribner Associates, USA). The test was operated at 80 °C under $H_2$ and $O_2$ condition with a backpressure of 0.2 MPa on the anode and cathode. $H_2$ and $O_2$ humidified at 80 °C (100% RH) were supplied to the anode and cathode compartments with a flow rate of 1000 sccm. The durability test was conducted with the flow rate of $H_2$ and $O_2$ being 300 and 500 sccm under otherwise identical conditions.

### Density functional theory calculations

The density functional theory (DFT) calculations were carried out using the Vienna Ab-initio Simulation Package (VASP)[71,72] with the projector augmented wave (PAW) method[73] and the Perdew-Burke-Ernzerhof (PBE)[74,75] exchange-correlation functional. Dispersion interactions were described using the DFT-D3 method proposed by Grimme[76,77]. The hetero-structure was created by the vaspkit code[78]. A $1 \times 1 \times 1$ k-point grid with a cutoff energy of 450 eV was used for the slab optimization and the density of states (DOS) calculation. Convergence was reached when the change in energy per electron step was less than $1 \times 10^{-5}$ eV. The conjugate gradient algorithm was used to relax all optimizable atoms until the force applied on them was less than 0.02 eV $Å^{-1}$ for the slabs.

The anatase $TiO_2$ (100) surface and $Ni_4Mo$ (211) surface were chosen for a better lattice match to create the hetero-structure using the slab model with a vacuum layer along the z-axis set to 15 Å. Two layers of $TiO_2$ (100) ($2\sqrt{5} \times \sqrt{7}$) surface were used as the substrate, and frozen during the optimization procedure, while the $Ni_4Mo$ (211) ($\sqrt{10} \times 3$) surface with a thickness of 6.15 Å above the $TiO_2$ substrate was cut by $\frac{1}{3}$ along the y-axis to expose the interface of the hetero-structure, which was allowed for optimization during the whole calculation. The lattice mismatch for this hetero-structure was kept as low as 0.011%.

Adsorbates were added on the interface of the hetero-structure. The binding energy of H atom and OH species was calculated using the flowing formula:

$$E_{H\,adsorb} = E_{slab+H} - E_{slab} - \frac{1}{2}E_{H2} \qquad (6)$$

$$E_{OH\,adsorb} = E_{slab+OH} - E_{slab} - E_{OH} \qquad (7)$$

Where $E_{H\,adsorb}$ and $E_{OH\,adsorb}$ are the adsorption energies of H atom and OH species, $E_{slab+H}$ and $E_{slab+OH}$ are the total energies of the compositions, $E_{H2}$ and $E_{OH}$ are the energies of hydrogen molecule and OH species, and $E_{slab}$ is the energy of the slab.

### Data availability

All data in the article and supplementary information are available from the corresponding authors upon request.

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

## Acknowledgements

The authors would like to thank the financial support from the National Natural Science Foundation of China. Grant numbers: 22172112 and 21773171 (S.W.C.); 21991150 and 21991154 (Z.L.). S.W.C. also thanks the financial support from the Fundamental Research Funds for the Central Universities.

## Author contributions

T.X.Y. and R.R.J.: experiments, data collection and analysis, writing the original draft; W.F.Y.: DFT calculations; P.J.J and Z.Z.B.: XAFS experiment and analysis; Z.L. and S.W.C: conceptualization, supervision, funding acquisition and editing the manuscript.

## Competing interests

The authors declare no competing interests.
