## [Peer Review File · Nature Communications]

REVIEWER COMMENTS

Reviewer #1 (Remarks to the Author):

First and foremost, I congratulate the authors with the nice and important results obtained. The study by Tian et al. entitled “Metal-Support Interaction Boosts the Stability of Ni-based Electrocatalyst to a New Record for Alkaline Hydrogen Oxidation” demonstrates important progress in the development of stable Ni-based catalysts for the anodes of AEMFC. The improved catalytic performance of Ni₄Mo/TiO₂ and Ni₂W/TiO₂ catalysts in the HOR is demonstrated and assigned to the metal-support interaction based on the thorough ex situ and in situ analysis of various materials by several physicochemical techniques. The achieved high power density and durability of the developed anodes in AEMFC tests further support the importance of the work. I hope the authors will clarify the comments outlined below to further improve the manuscript.

Major comments:

- If I understand right, the authors claim that metal Ni sites (needed for H adsorption) exist on the surface up to very high potentials if TiO₂ is added. At the same time, the CVs under the N₂ atmosphere (Fig. S23) show anodic peaks below 0.4 V, typically assigned with the formation of α -Ni(OH)₂, followed by the current decay. Could you please comment on this contradiction?
- According to the in situ XPS analysis, the percentage of Ni⁰ sites in Ni₄Mo and Ni₄Mo/TiO₂ stays roughly the same (within an error of the measurement) up to 0.5 V (Fig. S20d), while the HOR current decays already above 0.2V in the former case. Can you provide an explanation for this?
- I am surprised that the authors do not present a more detailed RDE analysis of the developed materials. It is highly recommended to demonstrate that the observed limiting current depends on the rotation rate, extraction of kinetic parameters will provide additional important information. In the latter context, I suggest modifying Table S2 by adding mass- and surface-normalized kinetic current densities at 50 mV for comparison with the literature.
- Though the choice of Ni₄Mo and Ni₂W active components for the investigation is well justified by the authors, I believe it is highly important to compare these systems with a more simple Ni/TiO₂ catalyst. Will the proposed metal-support interaction enhance the stability of Ni catalysts as well?
- As a follow-up to the previous question, can the authors provide more information about the stability of Mo and W in the studied catalysts against dissolution in the operated conditions? Recently, several studies (see, for example, ACS Catal. 2019, 9, 6837–6845; Electrochim. Acta 2018, 259, 1154–1161; ACS Catal. 2022, 12, 15341–15351) showed that both metals should dissolve at least from the surface of NPs.

The results of the Raman analysis provided in the text seem to be in line with these observations. Could the authors provide more details (based on XPS and TEM-EDS analysis) on how the concentration of metals (Ni, Mo, Ti, W) changes before and after the tests in RDE and MEA setups? Showing Mo3d spectra (acquired at different conditions) in the Supporting Information is highly recommended. In addition, I encourage the authors to perform an HR-TEM (EDS) analysis of the Ni₄Mo/TiO₂ catalyst after long-term electrochemical measurements to confirm the stability of the active component.

Minor major comments:

- Please provide in the Supporting Information more details about the in situ Raman and especially XPS measurements (such as the cell design, experimental protocol, etc.). In particular, I would like to know how the authors managed to perform in situ XPS analysis, which is typically done on synchrotrons, by using laboratory equipment. The quality of Raman spectra also seems very good, especially considering the fact that they were obtained without using the plasmonic effect (SERS). Providing more technical information will stimulate progress in the community.

Major minor comments:

- Please specify whether the potential was corrected for the ohmic contribution in both RDE and MEA studies. What was the typical ohmic resistance measured by EIS? Is the addition of TiO₂ affecting the value? Does this parameter depend on the applied potential?

- For Raman analysis (Fig. 5c), the authors write “the band at 460 cm⁻¹ starting from 0.1 VRHE ... attributed to the Ni-OH symmetric stretching mode of Ni(OH)₂”. First, I am not convinced that the band appears at 0.1V and not at ocp. It is hard to distinguish with the available signal-to-noise resolution. Besides, one can observe the same ‘band’ in Fig. 5d at least on spectra taken at 0.2 and 1.2 V. Do the authors strongly believe in the existence of the band at 460 cm⁻¹? Second, the band at 939 cm⁻¹ is a way more clear evidence for the Ni-O formation. It seems to appear already at 0.6 V, but then absent at 0.8 V. Can you please comment on this? Have the authors tried to record the spectra for both Ni₄Mo and Ni₄Mo/TiO₂ at higher potentials (in the Ni³⁺ region) to ensure the correctness of the analysis by visualizing highly intense bands assigned to NiOOH?

- The authors write “There is continuous HOR current at even higher potential (1.4 VRHE) before oxygen starts to evolve (data not shown).” Please show these data in the Supporting information along with CVs acquired in a wide potential region (at least up to 1.6 V vs RHE) for the most important samples (Ni₄Mo, Ni₄Mo/TiO₂, etc.).

Minor comments:

- I suggest modifying the title of the work by removing the phrase “a new record”, as this might sound inappropriate with time.

- Why Nafion was used for RDE studies and not QAPPT applied for MEA tests? Will the type of ionomer affect the performance of the catalysts?

- Please provide details of how the ECSA of Ni was calculated.

Reviewer #2 (Remarks to the Author):

This work reports a TiO₂ supported Ni₄Mo (Ni₄Mo/TiO₂) catalyst that can effectively catalyze HOR in alkaline electrolyte. The origin for the prominent activity and stability is attributed to the down-shifted d band center, caused by the efficient charge transfer from TiO₂ to Ni. This work is suggested to be published after addressing the following questions.

1. Ni₄Mo reaches the limiting current density of 2.65 mA cm⁻² geo, while Ni₄Mo/TiO₂ shows limiting current density of 2.23 mA cm⁻² geo.

What is the reason for the difference in limiting current density? What does such difference mean?

2. The HOR current of Ni₄Mo quickly drops at 0.2 VRHE, while Ni₄Mo/TiO₂ demonstrates extraordinary HOR activity even up to 1.0 VRHE, which is explained by the surface oxidation of Ni₄Mo which blocks the active surface areas and prohibits the hydrogen oxidation.

So the surface oxidation of Ni₄Mo is totally eliminated on Ni₄Mo/TiO₂? Any proof?

3. Ni₄Mo/TiO₂ demonstrates extraordinary HOR activity even up to 1.0 VRHE, with a mere 10% limiting current density decay.

What is the reason for such 10% limiting current density decay?

4. Figure 2, why Ni₄Mo/TiO₂ stops at 0.6V? and Ni₄Mo stops at even 0.7V? Fuel cell should stop at 0.1V or even lower potential.

5. The author explained the existence of the charge transfer from TiO₂ to Ni₄Mo by XPS, it is recommended to provide more proof such as EXAFS.

Reviewer #3 (Remarks to the Author):

I recommend the reconsideration of this paper after addressing following issues.

1. The current density and power density reported in this work are normalized by the mass of Ni metal. Is it appropriate? What is the role of Mo and Ti? Is it fair when compared with references as listed in Table S2 and S4.
2. The reported electrocatalyst shows obvious cell voltage degradation as shown in Figure 2b. What is the reason? Is there any morphology and composition change of the electrocatalyst.
3. The high electrocatalytic activity is attributed to the metal-support interactions. How about the performances of Ni₄Mo when loaded on other metal oxide supports? what is the special role of TiO₂ in the catalytic process?

REVIEWER COMMENTS

Reviewer #1 (Remarks to the Author):

First and foremost, I congratulate the authors with the nice and important results obtained. The study by Tian et al. entitled “Metal-Support Interaction Boosts the Stability of Ni-based Electrocatalyst to a New Record for Alkaline Hydrogen Oxidation” demonstrates important progress in the development of stable Ni-based catalysts for the anodes of AEMFC. The improved catalytic performance of Ni₄Mo/TiO₂ and Ni₂W/TiO₂ catalysts in the HOR is demonstrated and assigned to the metal-support interaction based on the thorough ex situ and in situ analysis of various materials by several physicochemical techniques. The achieved high power density and durability of the developed anodes in AEMFC tests further support the importance of the work. I hope the authors will clarify the comments outlined below to further improve the manuscript.

Response: We are very grateful for the reviewer’s high praise on the development of the efficient and stable Ni₄Mo/TiO₂ and Ni₂W/TiO₂ HOR catalysts reported in this work.

Major comments:

1. If I understand right, the authors claim that metal Ni sites (needed for H adsorption) exist on the surface up to very high potentials if TiO₂ is added. At the same time, the CVs under the N₂ atmosphere (Fig. S23) show anodic peaks below 0.4 V, typically assigned with the formation of α -Ni(OH)₂, followed by the current decay. Could you please comment on this contradiction?

Response: We thank the reviewer for this insightful comment and question. The cyclic voltammograms of Ni₄Mo and Ni₄Mo/TiO₂, collected in N₂-saturated 0.1 M NaOH, demonstrate the phase transformation of these catalysts (Figure R1a). The anodic current below ~0.3 V, typically assigned to the formation of α -Ni(OH)₂, appears on both Ni₄Mo and Ni₄Mo/TiO₂. This in situ formed α -Ni(OH)₂ can be reversibly (completely or partially) reduced to metallic Ni (Ni⁰) during the negative-going sweep or electrochemical activation at negative potentials, depending on the applied potential and the time. Accordingly, the surface of Ni-based materials is generally a combination of Ni⁰ and α -Ni(OH)₂ in the low HOR polarization region, and Ni⁰ is believed to

contribute to the HOR. On the positive-going sweep, Ni^0 in Ni_4Mo is gradually oxidized to $\alpha\text{-Ni}(\text{OH})_2$. Ni_4Mo exhibits very high HOR activity before 0.2 V due to its excellent intrinsic activity. When the potential is higher than 0.2 V, the surface of Ni_4Mo is fully oxidized to Ni (hydro-)oxide, making it deactivated for HOR. The polarization curve collected in H_2 eventually tracks the CV curve collected in N_2 (Figure R1b), suggesting that the surface oxidation of Ni_4Mo blocks the active surface areas and prohibits the hydrogen oxidation. Nevertheless, such surface oxidation to Ni (hydro-)oxide is significantly mitigated on the $\text{Ni}_4\text{Mo}/\text{TiO}_2$ catalyst, indicated by the considerably suppressed oxidation current in Figure R1a. Further in situ Raman study also suggest that the Ni-OH formation is much delayed until 1.0 V. Therefore, we think it is not contradictory, as the $\text{Ni}_4\text{Mo}/\text{TiO}_2$ has larger fraction of Ni^0 in a much wider potential window, which is responsible for its high HOR activity.

Figure R1. (a) Cyclic voltammograms of Ni_4Mo and $\text{Ni}_4\text{Mo}/\text{TiO}_2$ in N_2 -saturated 0.1 M NaOH at 20 mV s^{-1} . The potentials are not iR -corrected; and (b) positive-going sweeps of the cyclic voltammograms of Ni_4Mo and $\text{Ni}_4\text{Mo}/\text{TiO}_2$ recorded in H_2 and N_2 -saturated 0.1 M NaOH at 1600 r.p.m with a scanning rate of 0.5 mV s^{-1} of Ni_4Mo and $\text{Ni}_4\text{Mo}/\text{TiO}_2$. The potentials are iR -corrected.

We have updated Figure S23 in the original SI to Figure S29 in the revised SI with Figure R1a on page 35.

2. According to the in situ XPS analysis, the percentage of Ni^0 sites in Ni_4Mo and $\text{Ni}_4\text{Mo}/\text{TiO}_2$ stays roughly the same (within an error of the measurement) up to 0.5 V (Fig. S20d), while the HOR current decays already above 0.2 V in the former case. Can you provide an explanation for this?

Response: We appreciate the reviewer for the insightful comment and question. By plotting the Ni^0 percentage at different potentials with the HOR polarization curve, we expected to build the correlation between the HOR activity and the Ni^0 percentage. However, unlike the cyclic voltammetry method, which probes the real top surface of

the material, the XPS probes into a certain penetration depth. Although Ni₄Mo and Ni₄Mo/TiO₂ show significant difference in the surface property (see from the CVs in N₂, Figure R1), the difference in the Ni⁰ percentage might not be as significant as that in the electrochemical measurement, as the majority of Ni⁰ percentage may come from the few layers underneath the surface. Therefore, this correlation is only expected to qualitatively reveal that Ni⁰ is critical for the HOR. Considering that XPS is a semi-quantitative analysis technique, we therefore re-performed the in situ XPS to confirm the results.

The newly collected XPS spectra are shown in Figure R2. Starting from 0.3 V, the Ni⁰ percentage in the Ni₄Mo sample continues to decrease, while the Ni⁰ percentage in the Ni₄Mo/TiO₂ sample stays unchanged first, and then decreases at a slower rate (Figure R2d). Despite small difference in the absolute values, the trends in the change of Ni⁰ percentage are proven to be similar in the two sets of XPS measurements. Focusing on the trend in the change of Ni⁰, Ni²⁺ and Ti³⁺ percentage, we averaged the content data from the two measurements as shown in Figure R3, and updated Figures S26c, d on page 31 (Figures S20c,d in the original SI) and Figure S27b on page 32 (Figure S21b in the original SI) in the revised SI with Figure R3.

Figure R2. In situ Ni 2p_{3/2} XPS spectra of (a) Ni₄Mo and (b) Ni₄Mo/TiO₂ during the HOR at selected potentials

in 0.1 M NaOH; (c) Ni^{2+} contents estimated from XPS; and (d) Ni^0 ($\text{Ni}^{\delta-}$) contents estimated from XPS and the HOR polarization curves. The potentials are iR -corrected.

Figure R3. (a) Ni^{2+} , (b) Ni^0 and (c) Ti^{3+} contents estimated from the average of two sets of XPS measurements. The potentials are iR -corrected.

3. I am surprised that the authors do not present a more detailed RDE analysis of the developed materials. It is highly recommended to demonstrate that the observed limiting current depends on the rotation rate, extraction of kinetic parameters will provide additional important information. In the latter context, I suggest modifying Table S2 by adding mass- and surface-normalized kinetic current densities at 50 mV for comparison with the literature.

Response: We thank the reviewer for the thoughtful recommendation. Figures R4a and b show the HOR polarization curves on Ni_4Mo and $\text{Ni}_4\text{Mo}/\text{TiO}_2$ up to 1.0 V. Note that once the Ni_4Mo electrode was swept up to 1.0 V, the electrode would exhibit little HOR activity in the next scan due to detrimental surface oxidation. Even re-activation at negative potential (-0.1 V) would not recover the Ni_4Mo surface. $\text{Ni}_4\text{Mo}/\text{TiO}_2$ shows well-defined HOR limiting current up to 1.0 V at different rotation speeds. In order to obtain comparable kinetic current densities at 50 mV, the HOR polarization curves on $\text{Ni}_4\text{Mo}/\text{TiO}_2$ were also collected under the same condition as Ni_4Mo below 0.2 V (Figures R4c and d).

Figure R4. Positive-going sweeps of the HOR polarization curves of (a, c) Ni_4Mo and (b, d) $\text{Ni}_4\text{Mo}/\text{TiO}_2$ recorded in H_2 -saturated 0.1 M NaOH at various rotation speeds with a scanning rate of 0.5 mV s^{-1} . The insets show the Koutecky-Levich plots at 0.05 V. The potentials are iR -corrected.

We have updated the rotation speed dependent HOR polarization curves and the Koutecky-Levich plots of Ni_4Mo and $\text{Ni}_4\text{Mo}/\text{TiO}_2$ at 50 mV (Figure R4) as Figure S4 on page 6 in the revised SI. The mass normalized kinetic current densities at 50 mV are also added in Table S2 on page 10 in the revised SI. Besides, we have added the following paragraph on page 5 in the revised MS.

“Fig. S4 shows the HOR polarization curves of Ni_4Mo and $\text{Ni}_4\text{Mo}/\text{TiO}_2$ at different rotation speeds. Koutecky-Levich plots at 0.05 V exhibit a linear relationship between the inverse of i and $\omega^{1/2}$, with the slopes being 5.38 and $5.15 \text{ cm}^2 \text{ mA}^{-1} \text{ s}^{-1/2}$ for Ni_4Mo and $\text{Ni}_4\text{Mo}/\text{TiO}_2$ (insets of Figs. S4c and d). These values match reasonably well with the theoretical value of $4.87 \text{ cm}^2 \text{ mA}^{-1} \text{ s}^{-1/2}$ for the 2 e^- HOR (*J. Electrochem. Soc.* **2010**, 157, B1529-B1536), and are also in close agreement with the previous study (*Nat. Commun.* **2016**, 7, 10141).”

For the reviewer’s suggestion of adding surface normalized kinetic current density (specific activity) at 0.05 V in Table S2, as the reviewer also asked about how the ECSA was obtained (the last question from the reviewer), we re-consider the accuracy of ECSA measurement and the rationality of reporting specific activity, and decide not to report the specific activity. By providing the following explanation, we would like to reach an agreement with the reviewer.

Initially, the electrochemical surface area (ECSA) of catalyst was measured in N₂-saturated 0.1 M NaOH using the CV method from -0.3 to 0.6 V versus RHE with a rotating speed of 1600 r.p.m and a scanning rate of 50 mV s⁻¹. The OH desorption region on Ni was used to estimate the ECSA using a charge density of 514 μC cm⁻²_{Ni} for one monolayer of OH adsorption. The CV calculated ECSA of Ni₄Mo and Ni₄Mo/TiO₂ are 28.4 and 5.0 m² g⁻¹_{Ni} (Figure R5).

Figure R5. Cyclic voltammograms of Ni₄Mo and Ni₄Mo/TiO₂ recorded at 50 mV s⁻¹ in N₂-saturated 0.1 M NaOH. The potentials are not *iR*-corrected.

However, it has to be pointed out that although this method has been widely adopted to evaluate the ECSA of Ni-based catalysts, such as CoNiMo (*Energy Environ. Sci.* **2014**, 7, 1719-1724), Ni/N-CNT (*Nat. Commun.* **2016**, 7, 10141) and Ni-H₂-NH₃ (*Nat. Mater.* **2022**, 21, 804-810), it may not be very accurate for catalysts with complicated composites like Ni₄Mo/TiO₂ in our study, as our results suggest that OH adsorption behavior is quite different on Ni₄Mo/TiO₂. Accordingly, the conversion value of 514 μC cm⁻²_{Ni}, which assumes two OH on one Ni atom, may not be applicable.

Due to the lack of reliable methods to determine the ECSA of non-precious metals, it is difficult to compare specific activity with literatures. In our study, Ni₄Mo and Ni₄Mo/TiO₂ have similar mass activity. Considering that loading Ni₄Mo on to the TiO₂ support does not change the Ni₄Mo particle size, and there is partial coverage of Ni₄Mo in the presence of TiO₂, it is reasonable to rationalize that the specific activity of Ni₄Mo/TiO₂ is higher than that of Ni₄Mo, in line with the argument that the HBE on Ni₄Mo/TiO₂ is weaker than that on Ni₄Mo. This statement has been made on page 14 in the revised MS.

For a fair comparison, we emphasize the use of mass normalized activity for

performance evaluation and comparison with literatures. Wang *et. al* also highlight the use of mass activity instead of specific activity, due to the uncertainty in the ECSA determination (*Angew. Chem. Int. Ed.* **2021**, 60, 5771-5777). We therefore decide not to report the specific activity, and would like to delete the specific activity column in Table S2 on page 10 in the revised SI.

4. Though the choice of Ni_4Mo and Ni_2W active components for the investigation is well justified by the authors, I believe it is highly important to compare these systems with a more simple Ni/TiO_2 catalyst. Will the proposed metal-support interaction enhance the stability of Ni catalysts as well?

Response: We did also attempt to apply the metal support interaction (MSI) strategy to the Ni/TiO_2 system. The Ni/TiO_2 catalysts were synthesized using the same procedure as Ni_4Mo/TiO_2 and Ni_2W/TiO_2 without adding Mo and W precursors. The synthesis conditions were optimized by adjusting the Ti/Ni ratio (0.25, 0.17 and 0.13), annealing temperature (300, 400 and 500 °C) and annealing time (20, 60 and 120 mins). The HOR polarization curves of the best performing Ni/TiO_2 (Ti/Ni=0.17 and annealed at 400 °C for 60 mins), Ni (prepared under the same conditions without adding the TiO_2 support) and TiO_2 are shown in Figure R6. Because of the formation of Ni (hydro-)oxide, the Ni catalyst completely loses its HOR activity above 0.3 V. However, the addition of TiO_2 suppresses the OH adsorption on Ni, and thus keeps a stable and certain amount of HOR current density up to 1.0 V, suggesting the existence of the MSI.

Figure R6. Positive-going sweeps of the HOR polarization curves of Ni/TiO_2 , Ni and TiO_2 in H_2 -saturated 0.1 M NaOH at 1600 r.p.m with a scanning rate of 0.5 mV s^{-1} . The potentials are iR -corrected.

The MSI can improve the stability of Ni-based HOR catalysts, which is attributed to the down-shifted d band center, caused by the efficient charge transfer from TiO_2 to Ni.

The *d* band modification is expected to improve the HOR activity as well by tuning the HBE. However, the Ni itself has very poor HOR intrinsic activity owing to its too strong HBE (*ACS Catal.* **2018**, 8, 6665-6690; *Nat. Commun.* **2020**, 11, 4789; *Energy Environ. Sci.* **2014**, 7, 1719), and the enhancement in the HOR intrinsic activity induced by the MSI is thus very limited. This is why Ni/TiO₂ does not show satisfactory HOR activity, although its stability can be improved by the MSI.

5. As a follow-up to the previous question, can the authors provide more information about the stability of Mo and W in the studied catalysts against dissolution in the operated conditions? Recently, several studies (see, for example, ACS Catal. 2019, 9, 6837–6845; Electrochim. Acta 2018, 259, 1154–1161; ACS Catal. 2022, 12, 15341–15351) showed that both metals should dissolve at least from the surface of NPs. The results of the Raman analysis provided in the text seem to be in line with these observations. Could the authors provide more details (based on XPS and TEM-EDS analysis) on how the concentration of metals (Ni, Mo, Ti, W) changes before and after the tests in RDE and MEA setups? Showing Mo3d spectra (acquired at different conditions) in the Supporting Information is highly recommended. In addition, I encourage the authors to perform an HR-TEM (EDS) analysis of the Ni₄Mo/TiO₂ catalyst after long-term electrochemical measurements to confirm the stability of the active component.

Response: We thank the reviewer for the valuable comments and suggestions, and read very carefully the papers mentioned by the reviewer and also relevant ones (*ACS Catal.* **2022**, 12, 15341-15351; *Nat. Commun.* **2021**, 12, 2686; *ACS Catal.* **2019**, 9, 6837-6845; *Electrochim. Acta*, **2018**, 259, 1154-1161; *Angew. Chem. Int. Ed.* **2021**, 60, 7051-7055; *ACS Catal.* **2020**, 10, 12858-12866). We totally agree with the reviewer that both Mo and W elements suffer from intense dissolution from the surface of NPs due to the thermodynamic instability.

[Redacted]

Following the reviewer's comments, the electrochemical dissolution and stability of Ni₄Mo, Ni₄Mo/TiO₂, Ni₂W and Ni₂W/TiO₂ were systematically studied. Considering that XPS and EDS are semi-quantitative analysis methods, we also performed ICP-OES and ICP-MS to precisely study the Ni, Mo, W and Ti dissolution during the alkaline

HOR.

(1) The stability of Mo and W in the studied catalysts

I. Mo dissolution in the RDE study

We tested the Ni/Mo ratios of the relevant catalysts before the electrochemical measurement, after CV test at 0.5 mV s^{-1} from -0.05 to 1.0 V in H_2 -saturated 0.1 M NaOH (~ 30 mins), and after CA test at 1.2 V for 2 h in H_2 -saturated 0.1 M NaOH . The Ni/Mo ratios of Ni_4Mo and $\text{Ni}_4\text{Mo}/\text{TiO}_2$ after CV and CA tests are larger than the initial value of 4.0 (Table R1), indicating Mo dissolution during the electrochemical measurements. Assuming spherical shape of Ni_4Mo particles, Mo dissolution depth is roughly estimated to be ~ 0.5 - 1.1 nm from the surface of Ni_4Mo and $\text{Ni}_4\text{Mo}/\text{TiO}_2$, which is in consistence with the previous study (*Electrochim. Acta*, **2018**, 259, 1154-1161). ICP-MS results suggest that the Ni and Ti dissolutions are negligible. The TiO_2 support does not exhibit an inhibitive effect on Mo dissolution in the RDE study. Moreover, the Ni/Mo ratios after CV (~ 30 mins) and CA (2 h) tests remain roughly the same (ICP-OES), suggesting that the dissolution may happen at the beginning stage in the electrochemical measurements, which has also been observed in a similar study (*ACS Catal.* **2019**, 9, 6837-6845). Extension of time in electrochemical measurements may not further increase Mo dissolution. This is reasonable as Mo leaching from inner core is likely prevented by the Ni enriched surface formed during the reaction.

However, Mo dissolution does not seem to deteriorate the HOR performance of both Ni_4Mo and $\text{Ni}_4\text{Mo}/\text{TiO}_2$, which may be attributed to the following reasons: (1) Mo leaching is a surface effect, and the main active component remains as Ni_4Mo for HOR (see the post-reaction characterization presented in Figure R7); (2) selective Mo dissolution is found to increase the surface area (*Electrochim. Acta*, **2018**, 259, 1154-1161; *Mater. Sci. Eng. A*, **1997**, 226-228, 905-909); and (3) Mo in Ni_4Mo is oxidized and dissolved in the form of MoO_4^{2-} first, and the dissolved MoO_4^{2-} will polymerize and re-adsorb on the particle surface. The adsorption of $\text{Mo}_2\text{O}_7^{2-}$ dimer can promote the HER activity on Ni by down shifting the Ni d band center and accordingly weakening the H adsorption (*Angew. Chem. Int. Ed.* **2021**, 60, 7051-7055), which may also work for the HOR, as the H adsorption energy plays a decisive role in the HOR. Yet, the real mechanism behind it needs extensive study and is out of the scope of current study.

Table R1. Ni/Mo molar ratio and Mo dissolution depth of Ni₄Mo and Ni₄Mo/TiO₂ before and after the CV and CA tests.

Sample	Ni/Mo			Mo dissolution depth estimated from the ICP-OES result (nm)
	XPS	EDS	ICP-OES	
Ni ₄ Mo	10.4	4.3	4.1	—
Ni ₄ Mo after CV test	20.2	6.8	6.1	0.5
Ni ₄ Mo after CA test	14.5	10.8	6.9	0.6
Ni ₄ Mo/TiO ₂	7.9	4.8	4.4	—
Ni ₄ Mo/TiO ₂ after CV test	11.6	9.1	8.1	0.8
Ni ₄ Mo/TiO ₂ after CA test	18.5	14.2	10.6	1.1

Note: “Ni₄Mo” and “Ni₄Mo/TiO₂” samples represent the catalysts before the electrochemical measurements; “Ni₄Mo after CV test” and “Ni₄Mo/TiO₂ after CV test” samples represent the catalysts after CV tests in H₂-saturated 0.1 M NaOH from -0.05 to 1.0 V at 0.5 mV s⁻¹; “Ni₄Mo after CA test” and “Ni₄Mo/TiO₂ after CA test” samples represent the catalysts after CA tests at 1.2 V for 2 h in H₂-saturated 0.1 M NaOH. The potentials are not *iR*-corrected.

II. Mo dissolution in the MEA test

The Ni/Mo ratios of Ni₄Mo and Ni₄Mo/TiO₂ MEAs before and after AEMFC tests are shown in Table R2. The Ni/Mo ratios of Ni₄Mo and Ni₄Mo/TiO₂ MEAs before the AEMFC test are larger than 4.0, most likely due to the Mo dissolution during the MEA preparation process, wherein the catalyst-coated membrane had to be soaked in 1 M KOH at 60 °C for 24 h to exchange the anion of the membrane and ionomer to OH⁻. After AEMFC performance test, Mo in Ni₄Mo appears to dissolve with the Ni/Mo ratio reaching ~20. However, Mo exhibits an inconspicuous dissolution in Ni₄Mo/TiO₂ after AEMFC performance and durability tests.

Table R2. Ni/Mo molar ratio of Ni₄Mo and Ni₄Mo/TiO₂ MEAs before and after AEMFC performance and durability tests.

Sample	Ni/Mo	
	XPS	EDS
Ni ₄ Mo MEA before AEMFC performance test	6.2	6.8
Ni ₄ Mo MEA after AEMFC performance test	23.3	16.5
Ni ₄ Mo/TiO ₂ MEA before AEMFC performance test	6.9	7.3
Ni ₄ Mo/TiO ₂ MEA after AEMFC performance test	6.6	7.6
Ni ₄ Mo/TiO ₂ MEA after durability test	6.5	7.9

III. W dissolution in the RDE study

We also tested the W dissolution of Ni₂W and Ni₂W/TiO₂ under different conditions, and the results are similar to that of Ni₄Mo and Ni₄Mo/TiO₂ (Table R3).

Table R3. Ni/W molar ratio and W dissolution depth of Ni₂W and Ni₂W/TiO₂ before and after the CV and CA tests.

Sample	Ni/W			W dissolution depth estimated from the ICP-OES result (nm)
	XPS	EDS	ICP-OES	
Ni ₂ W	3.2	2.9	2.9	—
Ni ₂ W after CV test	5.2	2.3	3.6	1.1
Ni ₂ W o after CA test	4.6	4.4	3.7	1.2
Ni ₂ W/TiO ₂	2.9	2.0	2.4	—
Ni ₂ W/TiO ₂ after CV test	4.9	2.3	3.7	1.3
Ni ₂ W/TiO ₂ after CA test	3.8	2.4	3.7	1.3

Note: “Ni₂W” and “Ni₂W/TiO₂” samples represent the catalysts before the electrochemical measurements; “Ni₂W after CV test” and “Ni₂W/TiO₂ after CV test” samples represent the catalysts after CV tests in H₂-saturated 0.1 M NaOH from -0.05 to 1.0 V at 0.5 mV s⁻¹; “Ni₂W after CA test” and “Ni₂W/TiO₂ after CA test” samples represent the catalysts after CA tests at 1.2 V for 2 h in H₂-saturated 0.1 M NaOH. The potentials are not *iR*-corrected.

(2) The crystallographic, morphological and electronic structures of Ni₄Mo/TiO₂ after long-term electrochemical measurement

TEM, corresponding selected-area electron diffraction (SAED), EDS and XPS measurements were made to identify the active component of Ni₄Mo/TiO₂ catalyst after long-term stability test (CA at 1.2 V for 2 h).

1. The morphology, composition and crystal phases of Ni₄Mo/TiO₂ after long-term electrochemical measurement

TEM, HR-TEM and SAED measurements of Ni₄Mo/TiO₂ after long-term stability test (Figures R7a-c) demonstrate that Ni₄Mo/TiO₂ still has an interconnected particle morphology and the Ni₄Mo phase structure. The EDS mappings reveal that Ni, Mo, Ti and O elements also have a uniform spatial distribution (Figures 7d-h). The multiple post-reaction characterizations display that the morphology, composition and crystal phases are well maintained, clearly demonstrating the structural robustness of Ni₄Mo/TiO₂.

Figure R7. (a) TEM image, (b) HR-TEM image, (c) SAED pattern and (d-h) EDS elemental mappings of $\text{Ni}_4\text{Mo}/\text{TiO}_2$ after long-term stability test at 1.2 V (no iR -correction) for 2 h in H_2 -saturated 0.1 M NaOH.

II. The electronic structure of $\text{Ni}_4\text{Mo}/\text{TiO}_2$ after long-term electrochemical measurement

Figure R8 shows the Ni $2p_{3/2}$, Mo $3d$, Ti $2p$ and O $1s$ level XPS of $\text{Ni}_4\text{Mo}/\text{TiO}_2$ before and after long-term stability test. As demonstrated in Figure R8a, $\text{Ni}_4\text{Mo}/\text{TiO}_2$ after long-term stability test still has strong Ni^0 signals as the initial one. The Mo $3d$ level spectra show little Mo^0 signals after long-term stability test (Figure R8b), suggesting that Mo^0 species is not the HOR active component, and Mo dissolution does not affect the HOR performance. The Ti $2p$ and O $1s$ spectra in Figures R8c and d also exhibit an inconspicuous change after long-term stability test. The post-reaction XPS results indicate a robust electronic structure of the $\text{Ni}_4\text{Mo}/\text{TiO}_2$ catalyst.

Figure R8. (a) Ni $2p_{3/2}$, (b) Mo $3d$, (c) Ti $2p$ and (d) O $1s$ level XPS of $\text{Ni}_4\text{Mo}/\text{TiO}_2$ before and after long-term stability test at 1.2 V (no iR -correction) for 2 h in H_2 -saturated 0.1 M NaOH.

We have added a new section of “**Robust Structure of Ni₄Mo/TiO₂ in Electrochemical Measurements**” on page 13 in the revised MS. In this section, we included Figures R7 and 8 as Figures S19 and 20 on pages 23 and 24, Table R1 as Table S4 on page 25 in the revised SI, and discuss the stability of Ni₄Mo/TiO₂. Table R3 about W dissolution is added as Table S7 on page 44 in the revised SI, and discussion is provided on page 18 in the revised MS. Accordingly, we have updated the necessary references as [52-55] in the revised MS.

Minor major comments:

1. Please provide in the Supporting Information more details about the in situ Raman and especially XPS measurements (such as the cell design, experimental protocol, etc.). In particular, I would like to know how the authors managed to perform in situ XPS analysis, which is typically done on synchrotrons, by using laboratory equipment. The quality of Raman spectra also seems very good, especially considering the fact that they were obtained without using the plasmonic effect (SERS). Providing more technical information will stimulate progress in the community.

Response: We thank the reviewer for the suggestion. The experimental details for the in situ Raman and XPS are as follows:

“The in situ electrochemical Raman test was performed on a Renishaw Via Raman microscope with a 532 nm excitation wavelength and a 50× objective. The sample ink was prepared by dispersing the catalyst in a mixture of ultra-pure H₂O, isopropanol and Nafion, and then dropped on a piece of carbon paper (CP, 15×15 mm²), serving as the working electrode. Prior to collecting the Raman data, the Ni₄Mo and Ni₄Mo/TiO₂ working electrodes were scanned between -0.1 and 0.2 V versus RHE for 10 cycles in H₂-saturated 0.1 M NaOH in a home-made Raman cell (Figure R9), equipped with a platinum counter electrode and a SCE reference electrode. The working electrode was held at each potential (OCP and 0-1.5 V versus RHE at an interval of 0.1 V) for 2 mins for the Raman spectra collection. The laser intensity was set to be 10% with a collection time of 50 s for each spectrum.” The Raman spectra were obtained without using the SERS.

Figure R9. Schematic diagram of the in situ electrochemical Raman cell.

“Quasi in situ electrochemical XPS measurement was carried out on the ThermoFischer ESCALAB 250Xi instrument by applying a monochromatic Al K α X-ray source (1486.8 eV) operating at 12.5 kV and 16 mA under ultra-high vacuum (8×10^{-10} Pa). The total energy resolution was 0.10 eV. The catalyst ink was prepared by dispersing the catalyst in a mixture of ultra-pure H₂O, ethyl alcohol and Nafion, and then dropped on a glassy carbon (GC) electrode (4 mm in diameter), serving as the working electrode. Before collecting the XPS data, the Ni₄Mo and Ni₄Mo/TiO₂ working electrodes were scanned between -0.1 and 0.2 V versus RHE for 10 cycles in H₂/N₂ mixture-saturated 0.1 M NaOH in a home-made cell (Figure R10), equipped with a platinum counter electrode and a SCE reference electrode. The working electrode was first held at each potential (OCP, 0, 0.2, 0.3, 0.4, 0.5, 0.6, 0.8, 1.0 and 1.2 V versus RHE) for 2 mins, then vacuumed in the preparation chamber, and finally transferred to the test chamber for XPS spectrum collection without exposure to the air. All spectra were processed using the Shirley background correction, and calibrated with the C 1s component at 284.8 eV. The Gaussian-Lorentzian line shape was adopted to fit the spectra.”

Strictly speaking, as the XPS and the electrochemical measurements were not made simultaneously, i.e. the XPS spectrum was not collected while the electrochemical measurement was being performed, it was inappropriate to name the measurement as “in situ XPS”. We therefore change the term “in situ electrochemical XPS” to “quasi in situ electrochemical XPS”, which is also applied by Su *et al.* (*Nat. Comm.* **2022**, 13, 1322) in their study using the similar experimental procedures.

Figure R10. Schematic diagram of the quasi in situ electrochemical XPS cell.

We have included the above experimental details in the revised MS in **Methods** section on pages 25-26, and added Figures R9 and R10 as Figures S38 and S39 on pages 46 and 47 in the revised SI.

Major minor comments:

1. Please specify whether the potential was corrected for the ohmic contribution in both RDE and MEA studies. What was the typical ohmic resistance measured by EIS? Is the addition of TiO_2 affecting the value? Does this parameter depend on the applied potential?

Response: We thank the reviewer for the thoughtful suggestions and questions. In the RDE tests, the potentials were ohmic resistance corrected except for the stability test (CA), CV measurements in N_2 , and CO stripping measurements. We have made the differentiation by adding “The potential is iR -corrected”, “The potential is not iR -corrected” or “no iR -correction” in the figure captions and table notes in the revised MS and SI.

The potentials in the MEA studies were not ohmic resistance corrected, as AEMFC is a practical device and the practical performance is more concerned. Most of the AEMFC performances in previous studies are reported on the cell potentials with no ohmic resistance correction. The ohmic resistance corrected polarization curves were also provided in Figure R11 for comparison. We have updated the figure caption with “The cell voltage was recorded with no iR -correction.” in the original AEMFC Figure 2 on page 9 in the revised MS.

Figure R11. Polarization and power density curves of H₂/O₂ anion exchange membrane fuel cell (AEMFC) performance with Ni₄Mo (1.35 mg_{Ni} cm⁻²) and Ni₄Mo/TiO₂ (1.35 mg_{Ni} cm⁻²) at the anode and Pt/C (0.4 mg_{Pt} cm⁻²) at the cathode. (a) The cell voltages are not *iR*-corrected; and (b) the cell voltages are *iR*-corrected.

The EIS measurement was conducted to obtain the ohmic resistance (R) of the three electrode electrochemical system in the RDE study. As shown in Figure R12, the typical R values (the intersection of the Nyquist plot and the Z_{re} axis, marked with the black gridlines) of Ni₄Mo, Ni₄Mo/TiO₂ and TiO₂ electrochemical systems are 37, 38 and 38 Ω . The EIS measured R in the three electrode electrochemical system includes the solution resistance (R_s) between the work electrode (WE) and the reference electrode (RE), and the ohmic resistance of the WE. The R_s is intrinsically an ionic resistance, related with the electrolyte type and cell configuration. The measured 37-38 Ω is the typical solution resistance in an alkaline electrolyte (*J Electrochem. Soc.* **2010**, 157, B1529-B1536). The results suggest that the R_s is the dominant contribution in the measured R , and the TiO₂ addition has little effect on the R value of the system. Besides, the R values are independent on the applied potentials, as it is mainly composed of the solution resistance, which is related with the electrolyte type and cell configuration.

Figure R12. The Nyquist plots of the three electrode electrochemical system using (a) Ni₄Mo, (b) Ni₄Mo/TiO₂ and (c) TiO₂ as the working electrode. The data were collected in H₂-saturated 0.1 M NaOH at 1600 r.p.m at various potentials (no *iR*-correction).

2. For Raman analysis (Fig. 5c), the authors write “the band at 460 cm⁻¹ starting from 0.1 V_{RHE} ... attributed to the Ni-OH symmetric stretching mode of Ni(OH)₂”. First, I

am not convinced that the band appears at 0.1V and not at ocp. It is hard to distinguish with the available signal-to-noise resolution. Besides, one can observe the same 'band' in Fig. 5d at least on spectra taken at 0.2 and 1.2 V. Do the authors strongly believe in the existence of the band at 460 cm⁻¹? Second, the band at 939 cm⁻¹ is a way more clear evidence for the Ni-O formation. It seems to appear already at 0.6 V, but then absent at 0.8 V. Can you please comment on this? Have the authors tried to record the spectra for both Ni₄Mo and Ni₄Mo/TiO₂ at higher potentials (in the Ni³⁺ region) to ensure the correctness of the analysis by visualizing highly intense bands assigned to NiOOH?

Response: We thank the reviewer for the thoughtful comments and questions. First of all, the Raman spectra in the original MS may not be easy to see as ten lines were compressed in one plot. We accordingly replot the original spectra at selected potentials of interest to show the details (Figure R13). As shown in Figure R13a, the intensity of the band at 460 cm⁻¹ at 0.1 V is distinctly stronger than that at OCP, suggesting that Ni oxidation to Ni(OH)₂ likely appears at 0.1 V on Ni₄Mo. However, the same band disappears on Ni₄Mo/TiO₂ in the same potential window (Figure R13b). Figure R13c exhibits the spectra of Ni₄Mo/TiO₂ at 0.8, 1.0, 1.2 and 1.4 V, and it appears that the band at 460 cm⁻¹ seems to be slightly visible, starting from 1.0 V. The relative weak Raman signal is perhaps owing to the limited hydroxide thickness, as suggested in the previous study (*J. Electroanal. Chem.* **1992**, 333, 103-113). We conclude that Ni hydroxide forms from 0.1 V on Ni₄Mo, and the TiO₂ support inhibits the Ni surface oxidation to Ni(OH)₂ on Ni₄Mo/TiO₂ until a much higher potential of about 1.0 V.

Figure R13. In situ electrochemical Raman spectra of (a) Ni₄Mo and (b, c) Ni₄Mo/TiO₂ at selected potentials in 0.1 M NaOH during the HOR. The potentials are *iR*-corrected.

Second, we also replot the original spectra at selected potentials with a finer potential interval of 0.1 V to confirm the exact potential when the 939 cm⁻¹ band appears. As shown in Figure R14, the band of 893 cm⁻¹ at 0.6 V broadens with an inconspicuously

small hump at 916 cm^{-1} , which is about 23 cm^{-1} away from 939 cm^{-1} . As the resolution of Raman is 2 cm^{-1} , we do not consider the small hump at 916 cm^{-1} as part of the band at 939 cm^{-1} , and believe that the Mo-O peak starts from 0.9 V (1.0 V in the original MS) with the clear appearance of the 939 cm^{-1} band. However, we have no conclusion about the broadening of the band at 0.6 V , but tentatively ascribe it to the laser induced instantaneous local environment change of the Mo atoms in the MoO_4^{2-} tetrahedron.

Figure R14. In situ electrochemical Raman spectra of Ni_4Mo at selected potentials in 0.1 M NaOH during the HOR. The potentials are iR -corrected.

Finally, Figures 5c and d in the original MS only show the Raman spectra up to 1.2 V , but we did record the spectra of both Ni_4Mo and $\text{Ni}_4\text{Mo}/\text{TiO}_2$ up to 1.5 V (Figure R15). Although previous studies (*ACS Nano*, **2021**, 15, 13504-13515; *Angew. Chem. Int. Ed.* **2021**, 60, 19774-19778) suggest that the Ni-O bending (at 474 cm^{-1}) and stretching (at 558 cm^{-1}) vibration modes of $\gamma\text{-NiOOH}$ appear at $\sim 1.40\text{ V}$, we did not observe these two bands even at 1.5 V . We failed to obtain the Raman spectra at higher potentials due to the influence of the O_2 gas bubbles generated from the oxygen evolution reaction. To ensure the correctness of band assignment, the vibrational mode and Raman shift of possible Ni and Mo oxygenated species are summarized in Table R4, and updated as Table S5 on page 33 in the revised SI.

Figure R15. In situ electrochemical Raman spectra of (a) Ni_4Mo and (b) $\text{Ni}_4\text{Mo}/\text{TiO}_2$ at selected potentials in 0.1 M NaOH during the HOR. The potentials are iR -corrected.

Table R4. Vibrational mode and Raman shift of Ni and Mo oxygenated species.

Mode	Raman shift (cm^{-1})	Reference
Mo=O stretching mode in the MoO_4^{2-} tetrahedron	310	Angew. Chem. Int. Ed. 2021 , 60, 7051-7055
Mo=O bending mode in the MoO_4^{2-} tetrahedron	893	
bridging Mo-O-Mo symmetric stretching mode in $\text{Mo}_2\text{O}_7^{2-}$	483	Angew. Chem. Int. Ed. 2021 , 60, 7051-7055; J. Phys. Chem. C. 2010 , 114, 14110-14120;
		J. Phys. Chem. C. 1986 , 90, 6408-6411; J. Electrochem. Soc. 1988 , 135, 885-892; J. Electrochem. Soc. 2013 , 160, F235-F243; J. Electroanal. Chem. 1992 , 333, 103-113
Ni-OH symmetric stretching mode of $\text{Ni}(\text{OH})_2$	460	J. Phys. Chem. C. 2010 , 114, 14110-14120; J. Raman Spectrosc. 1990, 21, 683-691
Mo-O stretching mode of NiMoO_4	939	
Ni-O bending vibration mode of $\gamma\text{-NiOOH}$	474	ACS Nano , 2021 , 15, 13504-13515; Angew. Chem. Int. Ed. 2019 , 58, 17458-17464; J. Phys. Chem. C. 2012 , 116, 8394-840; Langmuir , 1998 , 14, 944-950
Ni-O stretching vibration mode of $\gamma\text{-NiOOH}$	558	

3. The authors write “There is continuous HOR current at even higher potential (1.4 V_{RHE}) before oxygen starts to evolve (data not shown).” Please show these data in the Supporting information along with CVs acquired in a wide potential region (at least up to 1.6 V vs RHE) for the most important samples (Ni_4Mo , $\text{Ni}_4\text{Mo}/\text{TiO}_2$, etc.).

Response: We thank the reviewer for the suggestions, and have added Figure R16a as Figure S2 (page 4), Figure R16b as Figure S3b (page 5) and Figure R16c as Figure S29 (page 35) in the revised SI.

Figure R16. (a) Positive-going sweeps of the cyclic voltammograms of Ni_4Mo and $\text{Ni}_4\text{Mo}/\text{TiO}_2$ recorded in H_2 and N_2 -saturated 0.1 M NaOH at 1600 r.p.m with a scanning rate of 0.5 mV s^{-1} . The potentials are iR -corrected; (b) chronoamperometry curves of $\text{Ni}_4\text{Mo}/\text{TiO}_2$ at 1.4 V (no iR -correction) in H_2 and N_2 -saturated 0.1 M NaOH at 1600 r.p.m; and (c) cyclic voltammograms of Ni_4Mo and $\text{Ni}_4\text{Mo}/\text{TiO}_2$ recorded at 20 mV s^{-1} in N_2 -saturated 0.1 M NaOH. The potentials are not iR -corrected.

Minor comments:

1. I suggest modifying the title of the work by removing the phrase “a new record”, as this might sound inappropriate with time.

Response: We appreciate the reviewer for pointing out this issue, and have modified the title to “Metal-support Interaction Boosts the Stability of Ni-based Electrocatalysts for Alkaline Hydrogen Oxidation”.

2. Why Nafion was used for RDE studies and not QAPPT applied for MEA tests? Will the type of ionomer affect the performance of the catalysts?

Response: We are grateful for the thoughtful questions by the reviewer. In the RDE study, Nafion is often used as the binder to fix the catalyst layer for its availability and convenience, which can be seen in the majority of current researches. Tiny amount of Nafion would not add additional resistance to the electrode kinetics. In our present study, preparing the catalyst ink by adding $50 \mu\text{L}$ Nafion (5 wt%, Sigma-Aldrich) into $200 \mu\text{L}$ H_2O and $750 \mu\text{L}$ isopropanol can successfully fix the catalyst layer on a glassy carbon electrode.

QAPPT is a commonly used ionomer for the MEA test. It needs to be dissolved in organic dimethyl sulfoxide (DMSO) solution, making the electrode preparation time-consuming as it takes a long time to dry in air. We tested the suitability of QAPPT (different mass fractions) as the binder for the RDE study. As shown in Figure R17, the

HOR performance is significantly suppressed with QAPPT as the binder, which may be originated from the influence of the remaining organic DMSO.

Figure R17. Positive-going sweeps of the HOR polarization curves of (a) Ni₄Mo and (b) Ni₄Mo/TiO₂ in H₂-saturated 0.1 M NaOH at 1600 r.p.m and 0.5 mV s⁻¹ with Nafion and various mass fractions of QAPPT as the binder. The potentials are *iR*-corrected.

3. Please provide details of how the ECSA of Ni was calculated.

Response: Please see our response to the reviewer’s 3rd question.

Reviewer #2 (Remarks to the Author):

This work reports a TiO₂ supported Ni₄Mo (Ni₄Mo/TiO₂) catalyst that can effectively catalyze HOR in alkaline electrolyte. The origin for the prominent activity and stability is attributed to the down-shifted d band center, caused by the efficient charge transfer from TiO₂ to Ni. This work is suggested to be published after addressing the following questions.

Response : We are very grateful for the reviewer’s highly positive opinions and support on the publication of this work.

1. Ni₄Mo reaches the limiting current density of 2.65 mA cm⁻²_{geo}, while Ni₄Mo/TiO₂ shows limiting current density of 2.23 mA cm⁻²_{geo}.

What is the reason for the difference in limiting current density? What does such difference mean?

Response: We thank the reviewer for this question. For the HOR test using the RDE technique, any catalyst would have the same diffusion limiting current (*i_L*) at a fixed rotation speed, according to the Levich equation: $i_L = 0.62nFAD^{2/3}\nu^{-1/6}c_0\omega^{1/2}$ (Bard, A. & Faulkner, L. *Electrochemical Methods: Fundamentals and Applications*, pp. 339),

which indicates that the HOR i_L depends solely on the rotation speed. However, it should be pointed out that the measured current is a superposition of HOR current, capacitive current and any oxidation/reduction current. The difference in the limiting current between Ni₄Mo and Ni₄Mo/TiO₂ mainly comes from the oxidation current and capacitive current, which are directly related with the catalyst loading amount.

To prove this, we performed another set of control experiments. Note that in the original MS, the Ni mass loading of Ni₄Mo/TiO₂ and Ni₄Mo are 376 and 477 $\mu\text{g cm}^{-2}_{\text{geo}}$ respectively; we then test the HOR on Ni₄Mo and Ni₄Mo/TiO₂ with the same Ni mass loading (376 $\mu\text{g cm}^{-2}_{\text{geo}}$). In the new set of data, Ni₄Mo reaches the limiting current density of 2.57 $\text{mA cm}^{-2}_{\text{geo}}$ at 90 mV, while Ni₄Mo/TiO₂ shows a limiting current density of 2.22 $\text{mA cm}^{-2}_{\text{geo}}$ at 90 mV (Figure R18). There is no significant change in the limiting current density (2.23 and 2.22 $\text{mA cm}^{-2}_{\text{geo}}$) of Ni₄Mo/TiO₂.

Figure R18. Positive-going sweeps of the cyclic voltammograms of Ni₄Mo and Ni₄Mo/TiO₂ in H₂ and N₂-saturated 0.1 M NaOH at 1600 r.p.m with a scanning rate of 0.5 mV s^{-1} . The potentials are iR -corrected.

Considering that TiO₂ has a very low specific capacitance (13.0 $\mu\text{F cm}^{-2}$ in the aqueous electrolyte, *J. Am. Chem. Soc.* **2018**, 140, 2397-2400) and CV in N₂ shows Ni₄Mo/TiO₂ has much less oxidation current, we assume that the observed limiting current density of 2.23/2.22 $\text{mA cm}^{-2}_{\text{geo}}$ is the pure HOR limiting current density. Therefore, as the Ni₄Mo mass loading decreases from 477 to 376 $\mu\text{g cm}^{-2}_{\text{geo}}$ (21% decrease), the contribution of oxidation current density and capacitive current density decreases from 0.42 (2.65-2.23) to 0.34 (2.57-2.23) $\text{mA cm}^{-2}_{\text{geo}}$ (19% decrease), proving that the difference in the limiting current density between Ni₄Mo and Ni₄Mo/TiO₂ mainly comes from the oxidation current and capacitive current of the catalyst.

2. The HOR current of Ni₄Mo quickly drops at 0.2 V_{RHE} , while Ni₄Mo/TiO₂ demonstrates

extraordinary HOR activity even up to 1.0 V_{RHE} , which is explained by the surface oxidation of Ni_4Mo which blocks the active surface areas and prohibits the hydrogen oxidation.

So the surface oxidation of Ni_4Mo is totally eliminated on Ni_4Mo/TiO_2 ? Any proof?

Response: We appreciate the reviewer for the questions. As claimed in the MS, the surface of Ni_4Mo is fully oxidized to Ni (hydro-)oxide when the potential is higher than 0.2 V, which blocks the active surface areas and prohibits the hydrogen oxidation. However, the surface oxidation of Ni_4Mo to (hydro-)oxide is significantly inhibited on Ni_4Mo/TiO_2 , leaving more clean metal Ni sites for HOR.

The inhibition of surface oxidation of Ni_4Mo to (hydro-)oxide on Ni_4Mo/TiO_2 can be proved by the CV plots and the in situ Raman spectra. As shown in Figure R19a, with the same Ni_4Mo mass loading, the oxidation current on Ni_4Mo/TiO_2 is significantly suppressed compared to Ni_4Mo . In addition, the Raman band at 460 cm^{-1} for Ni-OH symmetric stretching mode of $Ni(OH)_2$ appears at 0.1 V for Ni_4Mo (Figure R19b), indicating the formation of Ni hydroxide at 0.1 V during the HOR, while only when the potential reaches 1.0 V, this peak is marginally visible on Ni_4Mo/TiO_2 (Figure R19c).

Figure R19. (a) Cyclic voltammograms of Ni_4Mo and Ni_4Mo/TiO_2 in N_2 -saturated 0.1 M NaOH at 20 mV s^{-1} . The potentials are not iR -corrected; In situ Raman spectra of (b) Ni_4Mo and (c) Ni_4Mo/TiO_2 collected at the selected potentials in 0.1 M NaOH during the HOR. The potentials are iR -corrected.

3. Ni_4Mo/TiO_2 demonstrates extraordinary HOR activity even up to 1.0 V_{RHE} , with a mere 10% limiting current density decay.

What is the reason for such 10% limiting current density decay?

Response: We thank the reviewer for this question. In the CV test, the observed

current is the superposition of the pure HOR current, capacitive current and oxidation/reduction current. The capacitive current is minimized at a very slow scan rate of 0.5 mV s^{-1} , and the oxidation current is also very small. The observed slight decrease in the HOR polarization current is likely due to the surface oxidation induced activity decay. Although this surface oxidation on $\text{Ni}_4\text{Mo}/\text{TiO}_2$ is much reduced compared to Ni_4Mo , there is still a certain amount of surface (hydro-)oxide at higher potentials, which deactivates the HOR sites. We correlate the Ni^0 percentage from the quasi in situ XPS measurements with the HOR polarization curves, as shown in Figure R20. Though XPS is a semi-quantitative technique, this correlation qualitatively demonstrates that the Ni^0 percentage is critical to the HOR activity.

Figure R20. Ni^0 ($\text{Ni}^{\delta-}$) contents estimated from XPS and the HOR polarization curves. The potentials are iR -corrected.

4. Figure 2, why $\text{Ni}_4\text{Mo}/\text{TiO}_2$ stops at 0.6 V? and Ni_4Mo stops at even 0.7 V? Fuel cell should stop at 0.1 V or even lower potential.

Response: We appreciate the reviewer for the thoughtful comments and questions. With similar AEMFC assembly and test conditions, a lower cell voltage would usually lead to a higher polarization potential at the anode, which then requires an outstanding stability of the anode material to survive the harsh anodic condition. The metallic Ni, an oxyphilic non-precious metal, is not as stable as noble metals. Therefore, fuel cells with Ni-based anode catalyst usually fail to work steadily as that of noble materials to 0.1 V, because of the high anodic polarization induced catalyst oxidation, which has also been observed in previous studies (*J. Mater. Chem. A*, **2017**, 5, 24433-24443; *ACS Appl. Mater. Interfaces*, **2020**, 12, 31575-31581; *PNAS*, **2022**, 119, e2119883119; *J. Power Sources*, **2023**, 556, 232439).

For the Ni_4Mo anode AEMFC, the anodic Ni_4Mo catalyst would be irreversibly oxidized and deactivated when the cell voltage is below 0.7 V (0.685 V in Figure 2a in

the original MS), which is ascribed to its poor oxidation resistance. By adding the TiO₂ support, we improved the stability of Ni₄Mo/TiO₂ anode AEMFC to work at a lower cell voltage of ~0.6 V (0.543 V in Figure 2a in the original MS). It has to be mentioned that, even though the oxidation resistance of Ni₄Mo/TiO₂ is significantly improved, it is still not comparable to that of noble metals (Pt), which can work stably down to a cell voltage of 0.1 V. Developing highly active and stable non-precious HOR catalysts is the ultimate goal for AEMFCs.

5. The author explained the existence of the charge transfer from TiO₂ to Ni₄Mo by XPS, it is recommended to provide more proof such as EXAFS.

Response: We sincerely thank the reviewer for this suggestion.

The X-ray absorption fine structure spectroscopy (XAFS) is used to further probe the impact of the TiO₂ support on the chemical environment of Ni. Notably, the absorption edge of Ni₄Mo/TiO₂ displays a slight shift toward the lower photon energy relative to Ni₄Mo (Figure R21a, left inset), indicating the electron enrichment on Ni atoms in Ni₄Mo/TiO₂. The white line absorption intensity is also weaker than that of Ni₄Mo (Figure R21a, right inset), signifying the lower Ni valence state in Ni₄Mo/TiO₂ (Figure R21b). In addition, as shown in the Fourier-transform of Ni K-edge extended X-ray absorption fine structure (EXAFS) (Figure R21c), the intensity of the peak at 2.0 Å, assigned to Ni-Ni/Ni-Mo coordination of Ni₄Mo/TiO₂, is higher than that of Ni₄Mo, demonstrating an increased Ni coordination number, which is speculated to originate from the interaction with the TiO₂ support. The XAFS results confirm the MSI induced electron excursion from TiO₂ to Ni, which has also been observed in previously reported Ni@TiO_{2-x} catalyst (*ACS Catal.* **2017**, *7*, 7600-7609).

Figure R21. (a) Normalized Ni K-edge X-ray absorption spectra (the left inset is the magnified near-edge, and the right inset is the white line); (b) relation between the Ni K-edge absorption energy and valence state; and (c) Fourier-transform of Ni K-edge EXAFS spectra of Ni, NiO, Ni₄Mo and Ni₄Mo/TiO₂.

We have added these new data as Figures 4d and e on page 13 in the revised MS and

Figure S16 on page 20 in the revised SI. The above discussion and experimental details are provided on pages 11 and 21 in the revised MS.

Reviewer #3 (Remarks to the Author):

I recommend the reconsideration of this paper after addressing following issues.

Response: We greatly appreciate the reviewer's comments on our work.

1. The current density and power density reported in this work are normalized by the mass of Ni metal. Is it appropriate? What is the role of Mo and Ti? Is it fair when compared with references as listed in Table S2 and S4.

Response: We are grateful for the thoughtful comments and questions by the reviewer. We guess that the reviewer refers to Table S2 (current density) and Table S3 (power density), instead of Table S4, which is about the element ratio in the Ni₂W/TiO₂ catalyst. Moreover, in Table S3, the power densities are not normalized by the mass of Ni metal, and the Ni and Pt mass loadings are listed only for the reader's convenience. Now, the question is all about Table S2.

The best way to evaluate the intrinsic electrochemical activity of a catalyst is to extract the specific activity (SA) based on its accurate ECSA. However, owing to the lack of reliable methods to determine the ECSA of non-precious metals, it is difficult to obtain accurate SA and compare with the literature. From the perspective of practical application, we adopt the mass normalized activity (MA) in the present study for a universal comparison.

In terms of MA, there is also no consensus on how to report the HOR MA of Ni-based materials: the majority in the literature are based on the Ni loading amount only (63% in Table R5); some are based on the alloy loading amount (26% in Table R5); and very few are based on the catalyst loading amount (11% in Table R5). As the metallic Mo is HOR inactive (provides electronic effect in the alloy catalyst by regulating the Ni electronic structure to enhance the activity, *Nat. Commun.* **2020**, 11, 4789), and the surface Mo leaches from the nanoparticles under the experimental conditions (*ACS Catal.* **2019**, 9, 6837-6845), Mo is usually not considered to be involved in evaluating

the HOR MA. TiO₂ exhibits no alkaline HOR activity (Figure R22), and serves as the support to provide electronic effect for the improved stability. Therefore, we normalized the kinetic current by Ni mass, as it is generally considered that Ni provides active HOR sites. This argument is also applicable to Ni₂W and Ni₂W/TiO₂ catalysts.

Figure R22. Positive-going sweep of the cyclic voltammogram of TiO₂ in H₂-saturated 0.1 M NaOH at 1600 r.p.m and 0.5 mV s⁻¹. The potential is *iR*-corrected.

Following the reviewer's suggestion, we add various mass (Ni, Ni₄Mo metal and Ni₄Mo/TiO₂ catalyst) normalized HOR MA in Table R5, among which the Ni mass normalized MA is applied as the key parameter to evaluate the HOR performance in the present work. We also add the Ni₂W and Ni₂W/TiO₂ data in Table R5. We have updated Table R5 as Table S2 on page 10 in the revised SI.

Table R5. Experimental parameters and deactivation potentials of Ni-based non-precious metal electrocatalysts for the alkaline HOR.

Material	Loading (μg cm ⁻²)	Mass activity (A g ⁻¹)	Mass activity at 0.05 V (A g ⁻¹)	Scan rate (mV s ⁻¹)	Deactivation potential (V)	Electrolyte
Ni ₄ Mo/TiO ₂	376(Ni)	10.1(Ni)	29.6(Ni)	0.5	1.20	0.1M NaOH
	538(Ni+Mo)	7.1(Ni+Mo)	20.7(Ni+Mo)			
	874(cat)	4.3(cat)	12.7(cat)			
Ni ₄ Mo	477(Ni)	9.6(Ni)	26.2(Ni)	0.5	0.20	
	704(Ni+Mo)	6.5(Ni+Mo)	17.8(Ni+Mo)			
Ni ₂ W/TiO ₂	312(Ni)	6.8(Ni)	—	0.5	1.20	
	751(Ni+W)	2.8(Ni+W)				
	1060(cat)	2.0(cat)				
Ni ₂ W	349(Ni)	5.7(Ni)	—	0.5	0.23	
	817(Ni+W)	2.4(Ni+W)				
NiMo/KB	100(M+C)	4.5(M)	—	5	0.10	0.1M NaOH
Ni _{0.95} Cu _{0.05} /C	25(Ni+Cu)	2.5	—	2	0.15	0.1M NaOH
Ni/N-CNT	250(Ni)	3.5(Ni)	9.3(Ni)	1	Up to 0.08	0.1M KOH
CoNiMo	410(Ni)	5.0(Ni)	14.7(Ni)	Steady state	0.10	0.1 M KOH
Ni@h-BN	250(Ni)	3.5(Ni)	—	5	Up to 0.13	0.1M NaOH

Ni@C	100(Ni)	4.5(Ni)	—	10	Up to 0.10	0.1 M KOH
Ni ₃ N/C	160(Ni ₃ N)	12.0(cat)	24(cat)	1	0.26	0.1 M KOH
Ni/NiO/C	500 (cat)	—	5	1	Up to 0.10	0.1 M KOH
Ni/SC	138 (Ni)	7.4(Ni)	8.6(Ni)	5	Up to 0.10	0.1 M KOH
Ni/NC	167 (Ni)	4.8(Ni)	4.8(Ni)	5	Up to 0.08	0.1 M KOH
Ni/BC	168(Ni)	2.0(Ni)	2.2(Ni)	5	Up to 0.10	0.1 M KOH
np-Ni ₃ N	160(metal)	10.3	30	1	Up to 0.15	0.1 M KOH
Ni ₃ B/Ni	142(Ni)	7.0(Ni)	25(Ni)	5	Up to 0.11	0.1 M KOH
Ni ₄ Mo	200 (Ni+Mo)	14.1	54	1	0.20	0.1 M KOH
Ni ₃ N/Ni/NF	—	—	—	Steady state	Up to 0.10	0.1 M KOH
Ni ₄ Mo	500 (Ni+Mo)	6.8(metal)	68(metal)	0.5	Up to 0.20	0.1 M KOH
Ni ₄ W	500 (Ni+W)	3.7(metal)	17(metal)	0.5	Up to 0.20	0.1 M KOH
Ni/MoO ₂	765(cat)	—	—	5	Up to 0.10	0.1 M KOH
CeO ₂ /Ni	14(Ni)	7.6(Ni)	12(Ni)	5	Up to 0.11	0.1 M KOH
PS-MoNi	—	—	—	1	0.32	0.1 M KOH
Ni/MoO ₂	765(cat)	9.8(Ni)	39(Ni)	1	Up to 0.20	0.1 M KOH
Ni _{5.2} WCu _{2.2}	9200(cat)	2.5(Ni)	2.6(Ni)	1	0.30	0.1 M KOH
Ni/Ni ₃ N-C	166(Ni)	5.2(Ni)	12(Ni)	5	Up to 0.15	0.1 M KOH
4.3%N-Ni	320(cat)	9.3(cat)	77(cat)	1	0.25	0.1 M KOH
Ni@CN _x	500(cat)	—	1.2(Ni)	5	0.16	0.1 M KOH
Ni@O _i -Ni	142(Ni)	—	86(Ni)	5	0.26	0.1 M KOH
Ni ₅₂ Mo ₁₃ Nb ₃₅	8000 (metal)	—	—	1	0.80	0.1 M KOH

2. The reported electrocatalyst shows obvious cell voltage degradation as shown in Figure 2b. What is the reason? Is there any morphology and composition change of the electrocatalyst.

Response: We appreciate the reviewer for the questions. The cell voltage in Figure 2b in the original MS does show an obvious degradation after ~100 h (0.74 V to 0.50 V, ~32%). We are also curious about the origin of such a cell voltage decay.

Following the reviewer's considerate suggestion, we first performed multiple characterizations of the MEAs before and after durability test to study the change of the Ni₄Mo/TiO₂ catalyst. As demonstrated in Figure R23 (TEM, HR-TEM, SAED and EDS), Figure R24 (XPS), and Figure S7 in the original SI (XRD), the crystal, morphological, compositional, and electronic structures exhibit negligible difference before and after the durability test, revealing that the cell voltage degradation may not come from the anode catalyst side.

Figure R23. TEM images, HR-TEM images, SAED patterns and EDS elemental mappings of the $\text{Ni}_4\text{Mo}/\text{TiO}_2$ MEAs (a) before and (b) after the durability test.

Figure R24. (a) Ni $2p_{3/2}$, (b) Mo $3d$, (c) Ti $2p$ and (d) O $1s$ level XPS spectra of the $\text{Ni}_4\text{Mo}/\text{TiO}_2$ MEAs before and after the durability test.

Subsequently, we noticed the significantly increased high frequency resistance (ionic resistance) during the durability test (Figure R25), which is speculated to originate from the QAPPT ionomer (the adhesive and ionic conductor in the catalyst layer) degradation, as QAPPT is a kind of soft organic material and tends to degrade during the fuel cell

operation (*J. Power Sources*, **2018**, 390, 165-167; *RSC Adv.* **2017**, 7, 19153-19161; *Chem. Rev.* **2022**, 122, 6117-6321). Such degradation would limit the ion transportation and change the interface of the catalyst layer, resulting in an increased ionic resistance and then a dropped cell voltage. The non-negligible ionomer degradation may be an important source of the cell voltage degradation.

Figure R25. The high frequency resistance (HFR) of the fuel cell during the durability test.

However, the AEMFC cell voltage degradation is a systemic problem, which requires multifaceted consideration from the anode catalyst, cathode catalyst, ionomer and operation management. It is still a black box and impossible to quantify the contribution of each part through control experiments at current stage. It should be pointed out that the AEMFC with Ni₄Mo/TiO₂ anode catalyst operated at a high current density of 400 mA cm⁻² for ~100 h, exhibiting a better performance than the previously reported ones (~280 mA cm⁻² for 40 h in *Nat. Mater.* **2022**, 21, 804-810; 200 mA cm⁻² for 100 h in *PNAS*, **2022**, 119, e2119883119; 200 mA cm⁻² for 50 h in *Nat. Catal.* **2022**, 5, 993-1005; ~70 mA cm⁻² for 120 h in *ACS Appl. Mater. Interfaces*, **2020**, 12, 31575-31581). On the basis of above analysis, the Ni₄Mo/TiO₂ is by far the most stable AEMFC non-precious anode catalyst, which is expected to be competent for the AEMFC practical application.

We have added Figures R23 and 24 as Figures S22 and S23 on pages 27 and 28 in the revised SI, and provided discussion in the new section of “***Robust Structure of Ni₄Mo/TiO₂ in Electrochemical Measurements***” on page 13 in the revised MS.

3. The high electrocatalytic activity is attributed to the metal-support interactions. How about the performances of Ni₄Mo when loaded on other metal oxide supports? what is the special role of TiO₂ in the catalytic process?

Response: We are grateful for the thoughtful suggestions and questions by the

reviewer. Following the reviewer's suggestion, we applied the commonly used metal oxides in electrocatalysis (SnO_2 and CeO_2) to further explore the metal-support interaction (MSI). The HOR current densities of $\text{Ni}_4\text{Mo}/\text{SnO}_2$ and $\text{Ni}_4\text{Mo}/\text{CeO}_2$ with various Sn/Ni and Ce/Ni ratios significantly decay at ~ 0.2 V (Figure R26), indicating that the MSI strategy fails to work for $\text{Ni}_4\text{Mo}/\text{SnO}_2$ and $\text{Ni}_4\text{Mo}/\text{CeO}_2$ to improve the stability.

Figure R26. Positive-going sweeps of the HOR polarization curves of (a) $\text{Ni}_4\text{Mo}/\text{SnO}_2$ and (b) $\text{Ni}_4\text{Mo}/\text{CeO}_2$ in H_2 -saturated 0.1 M NaOH at 1600 r.p.m with a scanning rate of 0.5 mV s^{-1} . The potentials are iR -corrected.

Among the three studied metal oxide supports (TiO_2 , SnO_2 and CeO_2), TiO_2 seems to be particularly unique to improve the stability of Ni_4Mo , which has been attributed to the MSI between TiO_2 and Ni. This MSI also exists between TiO_2 and other transition metal, e.g., $\text{Pt}/\text{TiO}_x/\text{C}$ and $\text{Ru}@/\text{TiO}_2$ for HOR (*ACS Appl. Energy Mater.* **2019**, 2, 5534-5539; *Nat. Catal.* **2020**, 3, 454-462), Pd/TiO_2 for ORR (*J. Mater. Chem. A*, **2018**, 6, 2264), and also $\text{Ni}@/\text{TiO}_{2-x}$ for gas phase catalysis (*ACS Catal.* **2017**, 7, 7600-7609). TiO_2 may also play some other important roles in the metal-support composites, yet, more comprehensive and systematic research is needed to further explore the fundamental science behind it.

REVIEWERS' COMMENTS

Reviewer #1 (Remarks to the Author):

The authors have provided very detailed responses to my comments and have made several modifications to the manuscript that further enhance its high impact. Based on this, I am pleased to recommend this work for publication.

Reviewer #2 (Remarks to the Author):

The reviewer thinks this work can be accepted in current form.

Reviewer #3 (Remarks to the Author):

My concerns have been well addressed therefore I recommend its publication now.